

# Satellite telemetry of surface ablation observations to inform spatial melt modelling, Place Glacier, British Columbia, Canada

Alexandre R. Bevington[1,2], Brian Menounos[1,3,4], and Mark Ednie[5]

[1]Department of Geography, Earth and Environmental Sciences, University of Northern British Columbia, Prince George, BC, Canada
[2]Ministry of Forests, Province of British Columbia, Prince George, BC, Canada
[3]Geological Survey of Canada, Natural Resources Canada, Sidney, BC, Canada
[4]Hakai Institute, Campbell River, BC, Canada
[5]Geological Survey of Canada, Natural Resources Canada, Ottawa, ON, Canada

*Correspondence to*: Alexandre R. Bevington (alexandre.bevington@gov.bc.ca)

**Abstract.** Four automated "smart stakes" equipped with ultrasonic sensors, Arduino microcontrollers, and Iridium satellite telemetry were deployed to monitor glacier surface elevation changes at Place Glacier, British Columbia, Canada during the 2024 ablation season. The smart stakes recorded air temperature, relative humidity, and distance to glacier surface every 15 minutes from May 14 to September 21, 2024, providing high-temporal resolution melt data across an elevation gradient. Integration with airborne lidar surveys and satellite snow cover observations enabled validation and spatial extrapolation of point measurements. Temperature-index modeling using smart stake data yielded ice melt factors of –4.26 to –5.63 mm w.e. °C$^{-1}$ d$^{-1}$ and snow melt factors of –3.74 to –4.42 mm w.e. °C$^{-1}$ d$^{-1}$, consistent with previous studies. The spatial melt model estimated a total seasonal melt volume of $11.61 \times 10^6$ m³, representing a summer mass balance of –4.14 m w.e. for the glacier. Validation against manual ablation stakes showed reasonable agreement (R$^2$ = 0.58, RMSE = 0.45 m w.e.). Event–scale analysis revealed that three discrete heat events (July 5–22, August 1–12, and August 29–September 9) accounted for over half of the total seasonal melt despite comprising only one–third of the ablation season. Maximum daily melt rates reached –87 mm w.e. d$^{-1}$ during these extreme events, with higher elevation sites experiencing disproportionately greater melt rates. Non–linear temperature lapse rates were observed across the glacier, highlighting the importance of distributed temperature measurements for accurate melt modeling. The low–cost smart stake system demonstrates significant potential for automated glacier monitoring, providing near real–time data transmission and enabling event–scale melt attribution studies. This multi–scale monitoring approach combining in–situ sensors, airborne lidar, and satellite observations offers a comprehensive framework for understanding glacier melt dynamics in a changing climate, though challenges remain regarding sensor stability, power management, and accounting for glacier dynamics in melt estimates.



# 1    Introduction

Glacier mass balance is a key metric in understanding regional and global climate dynamics (IPCC, 2023). With a warming
climate and increases in summer heat waves, the process of ablation – which includes melting and sublimation of snow and
ice – is particularly important in determining a glacier's overall health (Cremona et al., 2023; Østrem, 1973; Pelto, 2019; Reyes
and Kramer, 2023). Ablation is typically quantified by surveying physical stakes that are drilled into glaciers (Glossary of
glacier mass balance and related terms). While these *in–situ* measurements are critical for constraining mass balance models
and understanding long term mass balance trends, these relatively infrequent observations are generally at a seasonal
resolution, for a given glacier, and limited to a small number of actively monitored sites (Hugonnet et al., 2021; Moore and
Demuth, 2001; Pelto, 2019).

In contrast, airborne and spaceborne remote sensing techniques are used to study mass balance over large areas (Clarke et al.,
2013; Hugonnet et al., 2021; Johnson et al., 2013; Kääb et al., 2012). Remote sensing can be used to, for example, map glaciers,
track snowline elevations, measure glacier velocities, and quantify volume changes (Bevington and Menounos, 2022, 2025;
Hugonnet et al., 2021; Lorrey et al., 2022; Østrem, 1973; Pelto, 2019; Rabatel et al., 2012). While airborne methods often have
a higher spatial resolution and vertical accuracy than spaceborne data, particularly with lidar derived elevation models, these
surveys can be expensive and few areas have repeat airborne lidar surveys over glaciers (Donahue et al., 2023; Pelto et al.,
2019). Despite their many advantages, remote sensing data have important limitations that are often mitigated with *in–situ*
observations (Podgórski et al., 2019). For example, the compromise between spatial and temporal resolution, geometric
distortions and shading issues in steep terrain, and the ongoing challenge of cloud cover. New methods are therefore emerging
for increased temporal sampling throughout the ablation season, that could allow further investigation of meteorological
forcing, event–scale data, and be an improved contribution for model testing (Cremona et al., 2023; Wickert et al., 2023).

Real–time data from glaciers is of increasing interest in order to inform real–time glacier melt and hydrological models
(Cremona et al., 2023; Wickert et al., 2023). Automatic weather stations are one solution to the need for high temporal
resolution real–time observations (e.g. Wheler and Flowers, 2011; Wickert et al., 2023). This equipment is generally costly
and can be complicated to install and maintain on a dynamic glacier surface. Due to the high cost, only one station is typically
used in combination with a network of traditional manual ablation stakes (e.g. Bash et al., 2018; Fitzpatrick et al., 2017).

These limitations have driven the glaciological community to test novel field techniques to study glacier change such as global
navigation satellite system interferometric reflectometry (GNSS–IR), repeat drone flights, low–cost ultrasonic sensors, and
cellular enabled timelapse cameras (Bash et al., 2018; Cremona et al., 2023; Landmann et al., 2021; Wells et al., 2024; Wickert
et al., 2023). However, in areas with no cellular network, few inexpensive telemetry options exist. Recent advances in low–
cost electronic microcontrollers (e.g. Horsburgh et al., 2019; Pearce et al., 2024) provide new avenues for satellite telemetry,
particularly through the use of short burst data satellite telemetry with polar orbiting satellites (Gomez et al., 2021). Low–cost





satellite solutions are well–suited for near real–time or moderate latency telemetry (e.g. minutes to hours) but less capable of real–time communication (e.g. minutes or less).

We report on the development and implementation of low–cost near real–time ablation stakes – herein referred to as 'smart stakes' that communicate outside of cell service. The objectives of the paper are:

1) Describe the design and performance of the smart stakes;

2) Compare the datasets to satellite and airborne remote sensing data; and

3) Demonstrate how real–time ablation data can be used to analyze melt attribution.

**Figure 1: A) Map of British Columbia highlighting the Fraser River watershed with the extent of panel B shown as a red square; B) Location of Place Glacier in relation to Pemberton and Lillooet Lake with a false–color shortwave infrared 2024 seasonal mosaic from Sentinel–2 satellite imagery as a basemap; D) Location of ablation stakes, smart stakes, and weather stations on Place Glacier with an August 2, 2024 orthoimage from aerial photography and lidar derived contour lines.**



## 2 Place Glacier

Place Glacier (RGI60–02.01104) is in the Southern Coast Mountains of British Columbia, approximately 20 km northeast of Pemberton (Figure 1). This gently sloping glacier has minimal crevassing and negligible surface velocities making it an ideal site for testing the smart stakes. Place Glacier is one of 61 reference glaciers worldwide and has one of the most complete mass

balance records in Canada (WGMS, 2024). Place Glacier has been the subject of multiple glaciology research studies (Donahue et al., 2023; Moore and Demuth, 2001; Mukherjee et al., 2023; Munro and Marosz-Wantuch, 2009; Richards and Moore, 2003; Shea et al., 2009; Wood et al., 2011).

Place Glacier is part of the Fraser River watershed and meltwater from the glacier flows into the Birkenhead River which enters Lillooet River before it enters Fraser River (Figure 1). The Place Creek watershed is about 14 km² and ranges in elevation

from 452 to 2588 m above sea–level (m asl). Place Glacier is the only glacier in this watershed and it decreased in area from 3.55 km² in 1985 to 2.53 km² in 2021, representing a 29% loss in 36 years (Bevington and Menounos, 2022).

## 3 Data and Methods

### 3.1 In–situ measurements

In this study, we use *in*-situ data from: 1) the new smart ablation stake network; 2) three automated weather stations; and 3) a

90 network of thirteen traditional mass balance stakes.

We installed four smart stakes towards the end of the accumulation season (May 8 and 9, 2024) that cover an elevation range from 1842 to 2156 m asl (Figure 1, Table 1). We accessed the glacier by helicopter, which adds financial considerations to the frequency of field visits. The uppermost region of the accumulation area was not instrumented due to field safety and logistical constraints. To prevent the stakes from melting out and tipping over, we redrilled the stakes on July 16, 2024, and again on

September 21, 2024. Site 1 is near the glacier terminus, where the glacier's surface slope is 10°. Site 2 is 535 m away on a gentler 9° slope with the same north aspect. Site 3 is on a flatter 5° northward flowing bench, right before the glacier turns westward up glacier. Site 4 is on a steep 16° east–northeast ramp that leads to a high elevation plateau (Figure 1, Table 1).

Three weather stations are used in this analysis (Figure 1, Table 1). The first, "Wx–Forefield", is a long–term weather station run by Natural Resources Canada located in the glacier forefield, some 400 m away from the glacier terminus. The second,

"Wx–Ridge", is a new weather station installed in early summer 2024 on an alpine ridge above glacier. The third, "Pemberton Airport CS", is a long–term Environment and Climate Change Canada weather station near Pemberton.

Data from thirteen manual ablation stakes collected by Natural Resources Canada as part of the WGMS are used in this study (Figure 1, Table 1). The stake positions and degree of melt are surveyed in autumn every year and snow depth and density is measured in the spring (WGMS, 2024). Field data collections for 2023–2024 occurred on October 7, 2023, April 19–20, 2024,

and September 20–21, 2024. Snow densities on April 19–20, 2024, ranged from 419 to 472 kg m$^{-3}$, with an average of 434 kg m$^{-3}$. The highest recorded densities were located at lower elevations on the glacier.





**Table 1: Metadata for the three weather stations, four smart stakes, and 13 WGMS stakes, see Figure 1.**

| Network | Name | Latitude, Longitude (WGS84) | Elevation (m asl) | Glacier Slope (°) | Glacier Aspect (°) |
|---|---|---|---|---|---|
| ECCC | Pemberton Airport CS | 50.3023, –122.7378 | 204 | – | – |
| NRCan | Wx – Forefield | 50.4323, –122.6079 | 1865 | – | – |
| VIU/UNBC/Hakai | Wx – Ridge | 50.4200, –122.6140 | 2306 | – | – |
| Smart Stakes | Site 1 | 50.4289, –122.6038 | 1842 | 10 | 347 |
| | Site 2 | 50.4242, –122.6022 | 1907 | 9 | 351 |
| | Site 3 | 50.4200, –122.6012 | 1975 | 5 | 8 |
| | Site 4 | 50.4165, –122.6092 | 2156 | 16 | 25 |
| NRCan/WGMS | #30 | 50.4283, –122.6031 | 1860 | – | – |
| | #35 | 50.4265, –122.6016 | 1886 | – | – |
| | #40 | 50.4240, –122.6002 | 1918 | – | – |
| | #45 | 50.4220, –122.5984 | 1957 | – | – |
| | #44 | 50.4203, –122.6040 | 1983 | – | – |
| | #50 | 50.4192, –122.5970 | 1987 | – | – |
| | #52 | 50.4175, –122.6003 | 2005 | – | – |
| | #75 | 50.4187, –122.6046 | 2028 | – | – |
| | #80 | 50.4172, –122.6071 | 2108 | – | – |
| | #90 | 50.4162, –122.6135 | 2220 | – | – |
| | #95 | 50.4144, –122.6139 | 2251 | – | – |
| | #100 | 50.4134, –122.6154 | 2274 | – | – |
| | #120 | 50.4119, –122.6193 | 2311 | – | – |

**Abbreviations**: ECCC: Environment and Climate Change Canada; NRCan: Natural Resources Canada; VIU: Vancouver Island University; Hakai: Hakai Institute; UNBC: University of Northern British Columbia; WGMS: World Glacier Monitoring Service; Wx: Weather Station.

### 3.2    Smart ablation stakes design

Our design priorities for the 'smart stakes' include: 1) low power consumption; 2) multiple sensor compatibility; 3) reliable satellite telemetry; 4) small size and mass; and 5) low total cost.

The smart stakes use an Arduino compatible Adafruit © Feather M0 Adalogger microcontroller, hereafter referred to as the M0 (Figure 2). Arduino is an open–source electronics platform based on easy–to–use hardware and software, widely used for prototyping interactive projects and embedded systems (Arduino, 2025). The M0 runs the ATSAMD21G18 ARM Cortex M0 processor, clocked at 48 MHz and has 3.3V logic (Adafruit Feather M0 Adalogger, 2025). The M0 is compatible with multiple analog and digital communication protocols (e.g. I2C, Analog, SDI–12, etc.), has a built–in micro–SD card reader, can be powered via USB or battery, and contains both FLASH (256K) and RAM (32K) memory. The data logger includes a screw terminal breakout board for the M0, a real–time clock, solar charger, power management chip, and the satellite modem (Figure 2, Table S1).

The power consumption of the Arduino system depends on the tasks assigned (e.g. waking up, time check, powering up and reading sensors, reading and writing data, and sending messages). In between measurements, the M0 can achieve low power consumption via a 'deep sleep' command that is in the range of ~4 mA (SleepyDog: Arduino library to use the watchdog timer for system reset and low power sleep, 2025). Our smart stakes, however, do not use deep sleep commands, as we found that



the power consumption using these libraries was still too high, and that the libraries would often fail over long periods of time. We instead incorporate a SparkFun TPL5110 Nano Power Timer into the design, this board serves as an intermediary between the battery and the M0. The TPL5110 turns off the system, including the M0, between measurements which reduces the sleep consumption to ~35 nA (3.5 x 10$^{-5}$ mA), and ensures that the entire Arduino system is reset for every measurement, which mitigates the issues of Arduino failure over long periods of time. In a case where the M0 wakes up every 15 minutes for 10 seconds to take a measurement at ~90 mA and communicates every 2 hours at ~245 mA for 2 minutes, the non–TPL version would consume 215 mAh of power per day, whereas the TPL version would only use 120 mAh. This difference in power consumption translates to usable battery life of about 47 and 82 days respectively, with 10,000 mAh Lithium Polymer (LiPo) battery and no solar panel. One disadvantage of the TPL chip is that the samples are taken at approximately 15 minute intervals.

The smart stakes have a small 6 V, 2 W solar panel which charges a 10,000 mAh 3.7 V lithium polymer (LiPo) battery. The charging of LiPo batteries is restricted by both cold (<0°C) and warm temperatures (>40°C). If the data logger runs out of power, the smart stake will stop working. In cases where the battery starts to charge again (temperatures warm, solar panel becomes snow–free), the TPL5110 is designed to restart the smart stake and operations will resume. This added feature adds resilience to extended periods with little or no charging in the winter.

### 3.2.1 Sensors

The smart stakes are equipped with ultrasonic distance, air temperature and relative humidity sensors (Figure 2). The ultrasonic sensor is the MaxBotix MB7374 HRXL–MaxSonar–WRST7, which is an inexpensive high precision range–finder that operates on low–voltages. It is specifically optimized for measuring snow and has recently been used in other glaciological studies (Wickert et al., 2023). Since the speed of sound increases by about 0.6 m s$^{-1}$ °C$^{-1}$, the ultrasonic has temperature compensation built into the sensor (HRXL-MaxSonar-WRS Datasheet, 2025). MB7374 can measure distances between 0.5 and 5 m. Factory data sheets report reading–to–reading stability of 1 mm at 1 m, and an overall accuracy of 1% or better. The sensor operating temperature ranges from –40°C to +65°C and the input voltage requirement is from 2.7 to 5.5 V.

The built–in temperature compensation resides inside the ultrasonic casing, which heats up in the sun. This sensitivity can manifest as variable distance measurements depending on the solar heating of the sensor. Typically, this can be corrected with independent air temperature measurements, however the MB7374 only reports the corrected distance, not the raw time of flight data and thus we cannot correct the data in post processing. The sensor can use a separate air temperature sensor to correct for this, however this can only be done using a dedicated air temperature sensor that is soldered directly to the ultrasonic and this option was not available to us during the time of study. We do, however, include a separate DFRobot SEN0148 temperature and humidity sensor that uses the Sensirion SHT31–ARP chip. It is a fully calibrated, linearized, and temperature compensated analog output with input voltage requirements from 2.4 to 5.5 V. Operating temperatures are between –40 and +125°C. Typical reported accuracy is 2%RH and ±0.3°C. The accuracy decreases to ±1.3°C at the limits of the temperature range.







**Figure 2: Photographs of the stations: A) Station wiring with M0 (Feather M0), RB (RockBlock 9603), RTC (Real–time Clock), TPL (TPL5110 Nano Power Timer), CC (Charge Controller), and battery (behind the wooden board); B) Place 2 on 2024–07–16; C) Place 2 on 2024–05–09; D) Place 1 on 2024–07–16; E) Place 2 on 2024–07–16; F) MaxBotix Ultrasonic Sensor (MB7374 HRXL–MaxSonar–WRST7).**



### 3.2.2    Telemetry

Local low frequency radio, cell networks, and satellite networks are options for sending data from the field (Cremona et al., 2023; Kodali, 2017). Due to the remote location of our study area, we use satellite telemetry. Satellite options include either geostationary (e.g. GOES) or low earth polar–orbiting satellites (e.g. Iridium). Geostationary satellites offer a low financial cost per message; however, the upfront cost of the modem is high. Geostationary satellite telemetry also requires the modem to be directly pointed at the satellite which can be a challenge for fast–flowing glaciers or those situated within rugged

topography, or in situations where the smart stake could slowly tip over while melting out. We use the Iridium satellite constellation due to the short wait times for satellite connectivity, flexible communication, and relatively low cost hardware (Gomez et al., 2021; IridiumSBD v2.0, 2024). The downside to Iridium is the cost per message structure and often 2–5–minute wait times for messages to successfully transmit.

Messages are sent from the smart stakes to GroundControl©, a commercial service that brokers Iridium messages. We forward

compressed messages as HTTP POST to a PostgreSQL database on a DigitalOcean© Server. Monthly line rentals are roughly $16 USD per unit, and the number of credits used per message varies depending on the size of the message. Credits vary in cost based on purchase volume from $0.06 to $0.15 each. One credit is used per 50 bytes of data, and the maximum message size is 340 bytes. Two hours of hourly data from our stations translates to about 47 bytes which, in our case, results in a total monthly transmission cost of $36 USD per smart stake. The can be queried and visualized using open–source software

(Northern BC Hydrology Research, 2025; Chang et al., 2025).

### 3.2.3    Installation

The ablation pole is made of two 2.44 m sections of 1–inch aluminum pipe joined by an aluminum coupler for a total length of 4.88 m. A 1.2 m length cross arm of the same pipe is then mounted near the top using readily available pipe couplers. The poles are then drilled into the glacier so that the ultrasonic sensor is ~0.8 m above the glacier surface. For Sites 1–3, the

snowpack was thin enough during the initial installation to drill the poles into the underlying ice, but thick snow at Site 4 prevented drilling the poles into the underlying ice.

The data logger is placed inside a Pelican 1120 Protective Case (Interior L×W×D = 18.5 x 12.1 x 8.5 cm) and the solar panel is directly mounted to the case using a large metal hose clamp. The case is then hooked onto the cross arm of the ablation stake using its handle and secured in place with a hose clamp. The ultrasonic sensor has a 19 mm thread and is mounted to a 90–

degree electrical box, which is mounted to the aluminum cross arm (Figure 2). To limit the edge effects of the ablation stake in the footprint of the sensor, we positioned the ultrasonic sensor about 0.8 m away from the ablation stake. The T/RH sensor is inside a radiation shield that is secured to the top of the main ablation stake.





### 3.3 Data cleaning

#### 3.3.1 Smart stake measurements

The time series of glacier surface observations recorded by the smart stakes require filtering, gap filling, and datum corrections. The ultrasonic sensor reports the maximum range of the sensor (5 m) if there is signal interference during precipitation events, drifting snow, or high winds. Objects closer than the minimum detection range (e.g. rime ice build–up on the sensor, spiderwebs, or other obstructions) are reported as 0.5 m. Bad data can also be caused by the stakes themselves leaning over or turning (described below). We infill missing data for periods of less than 5 hours using linear interpolation. The distance values

are converted to cumulative elevation change by adjusting the timeseries after each time the stake is redrilled, and the cumulative elevation change is corrected to elevation in meters above sea level (m a.s.l.) using airborne lidar and manual differential GPS measurements (described below). In addition, we determine when each station became snow–free using satellite imagery (described below).

#### 3.3.2 Airborne lidar elevation models

Airborne lidar and air photo data is available from May 11, June 6, July 4, August 2, August 31, September 16, and October 12, 2024. These flights are part of the Airborne Coastal Observatory glacier monitoring program (Donahue et al., 2023; Menounos et al., 2019). These acquisition dates overlap with the smart stake deployments, and the elevations can be compared directly to the smart stake data. Slope and aspect were calculated using the August 31, 2024 digital elevation model (DEM) in QGIS 3.34 (QGIS Development Team, 2023).

The DEMs were bilinearly resampled to a 1 m grid and co–registration of the DEM stack was done using the 'xdem' python package with stable exposed and snow–free bedrock nunataks (Dehecq et al., 2022). We identified bedrock nunataks in a May 11, 2024 air photo and assumed they were stable throughout the time series. Stable ground was used to determine the three–dimensional correction vectors. This was done with methods from Nuth and Kääb (2011) that estimate horizontal and vertical translations by iterative slope and aspect alignment. In addition, a 2D polynomial correction was applied to account for any

model deformations.

#### 3.3.3 Differential GPS

Since there is no stable ground near the smart stakes themselves, a relative datum could not be established through traditional survey methods. As such, the most reliable way to ensure the accuracy of the smart stakes is by correcting the data to a geodetic datum using differential GPS (dGPS) measurements. We have high resolution dGPS measurements from September 21, 2024,

for Sites 3 and 4. We use the Emlid Reach RS2+ dGPS system, that produced spatial accuracies of 0.006 m and vertical accuracy of 0.01 m. Points were recorded for the upon arrival and after resetting the stake.



### 3.3.4    PlanetScope snow cover

The PlanetScope Doves are a constellation of commercial CubeSat satellites that offer near–daily 3 m resolution optical imagery and have been used in other glaciological studies (Liu et al., 2024; Tarca et al., 2023). These data are used to investigate snowline retreat during the 2024 ablation season on Place Glacier and relate the snow cover to smart stake measurements. The 'planetR' package was used to bulk download scenes that intersect the bounding box of the area of interest (Bevington, 2023). Seventy–three orthorectified surface reflectance images had less than 20% cloud cover between May 1 and Oct 15, 2024.  We manually remove images that are not suitable for analysis due to clouds, haze, or partial coverage. We use the near infrared (NIR) band with a threshold of 0.4 to differentiate between snow and ice at our monitoring stations (Riggs et al., 1994; Zhang et al., 2019). This simple band threshold method is sufficient for discrimination of snow from glacier ice; however, it would not work well for off glacier areas, for supraglacial lakes, or when clouds are present.

### 3.4    Temperature–index model

### 3.4.1    Per–stake melt model

We use our *in–situ* data with a temperature–index melt model to estimate glacier melt from air temperature (Beedle et al., 2014; Braithwaite and Zhang, 2000; Carenzo et al., 2009; Pellicciotti et al., 2005; Shea et al., 2009; Wickert et al., 2023). This model operates on the simple assumption that air temperature is well correlated with shortwave radiation and is the main driver of melt, and that melt occurs when air temperature exceeds a threshold value, commonly set at 0°C. The amount of melt is then quantified using degree–day factors for ice and snow, which represent the amount of melt per degree Celsius per day, commonly expressed in meters of water equivalent per degree Celsius per day (m w.e. °C$^{-1}$ d$^{-1}$). The temperature–index model (Braithwaite and Zhang, 2000) can be expressed as in (1):

$$M = k_i \times PDD_i + k_s \times PDD_s \qquad \textbf{(1)}$$

where M is melt (m w.e.), $k_i$ and $k_s$ are the melt factors (m w.e. °C$^{-1}$ d$^{-1}$) for ice ($i$) and snow ($s$), and $PDD_i$ and $PDD_s$ are the positive degree days in degrees Celsius. This model is simple and allows an initial estimate of melt and does not account for more complex energy balance processes like incoming solar radiation, wind, humidity or albedo.

We convert surface elevation change measured at each smart stake to water volume using field observations of snow densities (spring 457±50 kg m$^{-3}$; summer 570±20 kg m$^{-3}$) and ice densities (910±10 kg m$^{-3}$) reported by Pelto et al. (2019). For this model, we only use the summer snow and ice densities from Pelto et al. (2019).

### 3.4.2    Spatial melt model

We combine lidar DEMs, smart stake air temperature, remotely sensed snow masks, and the snow and ice melt factors to estimate the total summer melt volume from Place Glacier for 2024 using (1. Seven lidar DEMs are bilinearly resampled to 5 m resolution and interpolated to daily resolution from May 11 to September 21, 2024 using a cubic spline pixel–wise time–series function (Figure S4). A daily gridded air temperature model is then estimated using linear air temperature lapse rates for





each day from the smart stakes and weather stations. This results in a 5 m resolution daily PDD dataset (Figure S5). Snow and ice cover maps are interpolated to a daily resolution from PlanetScope by gap filling the sparse timeseries. As this is a binary classification of snow or ice, we use the last good measurement from spaceborne observations to infill the missing daily values (Figure S6). The average observed melt factors for snow and ice at the smart stakes are then used to estimate the daily spatial distribution of melt. The daily gridded PDD is then multiplied with the daily melt factor distribution based on the presence of snow or ice, and results in a daily melt raster (Figure S7), which can be converted to a total seasonal melt volume (m$^3$) by multiplying the estimated melt (m w.e.) by the pixel resolution (in m) and adding together the daily totals. The summer balance can then be calculated as the melt volume divided by the glacier area. An example of the spatial melt model results is shown in Figure S8 for September 14, 2024.

## 4 Results

### 4.1 Smart stake performance

The smart stakes performed without fault for Sites 1 and 3 (Figure 1). Measurements of air temperature, relative humidity, and the distance to the glacier surface were recorded every 15 minutes and only the hourly data were sent over the Iridium satellite network. Site 2 had intermittent satellite communication for the beginning of the season, which we believe was caused by loose wiring, but the data were preserved on the local SD card. Site 2's intermittent communication corrected itself mid–summer, but the stake stopped transmitting after our last field visit of the season on September 25, 2024. The cause of this issue remains unknown as the logger is on the glacier at the time of writing. Site 4 performed well for the ablation season but then suddenly stopped working on September 11, 2024, due to a corrupt memory card.

We redrilled the smart stakes on July 16, and on September 21 to prevent them from melting out. On the September 21 field visit, Sites 1–4 were found to be slowly tipping over at angles of 41°, 31°, 24°, and 15°, respectively (Figure 3); the day where tilting commenced remains uncertain. As a precaution, we remove data from late–August to September 21 when the stakes recorded false accumulations while air temperatures were positive.

The MB7374 ultrasonic sensor performed well against *in–situ* calibrations (Figure S1). The root mean square error (RMSE) between field measurements done with an avalanche probe and the sensor itself had an overall error of 0.046 m. The avalanche probe used had a 1 cm graduation, and the highest values (>3m) have a larger uncertainty due to the challenge of seeing the exact measurement on the avalanche probe. The glacier surface also presents a challenge as it is a sloped and uneven surface. The MB7374 sensor was susceptible to solar heating, with an observed diurnal fluctuation of ±5.5 cm on hot sunny days (Figure S2). This could be mitigated in the future by using the external temperature sensor correction inputs that are built into the sensor's functionality. To our knowledge it is not possible to do this after the fact with the MB7374 using the air temperature that we recorded because the sensor only reports distance. It does not report the time of flight, or the temperature compensation used. The nighttime temperatures are likely the most reliable due to the absence of solar heating of the sensor and the temperature compensation built into the ultrasonic should be closer to the actual air temperature.





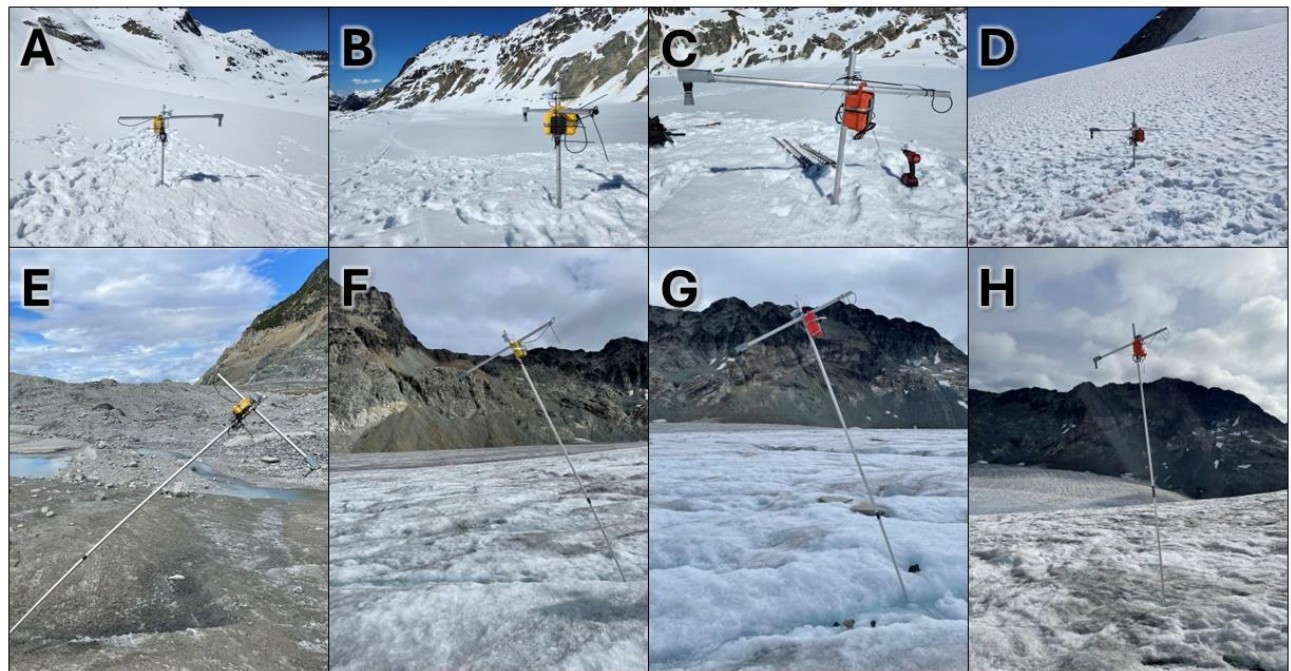

**Figure 3: Site photos from left to right (Sites 1–4). Photos were taken on July 16 after being redrilled (A, B, C, D), and on Sep 21 when found melting over before being re–drilled or removed (E, F, G, H).**

## 4.2    Satellite derived snow cover

For Sites 1–4, the start of the snow free season occurred on July 12, July 14, July 19, and August 15, 2024, respectively (Figure

4). Clouds during mid–August and mid–September introduced gaps into the satellite image time series. The transient snowline gradually rose over the ablation season with a summer snowfall event observed on August 28, 2024 (Figure 4). The accumulation area completely disappeared by September 24, 2024, and snowfall events became more frequent in October.







**Figure 4: A) Selected PlanetScope Dove near infrared scenes. The red outline is the snow mask determined from a 0.4 threshold of the near infrared band. B) Spectral time series of the NIR bands from PlanetScope for the smart stake locations. The black squares represent the first snow–free observation at that location.**





## 4.3    Time series data

The cumulative elevation change measured at each stake was adjusted to elevation above sea level using the June 6 lidar

(Figure 5). The RMSE between the lidar elevation and the valid smart stake measurements is 0.18, 0.11, 0.12, and 0.50 m for Sites 1–4, respectively. The largest differences are at Site 4, which may be due to the steeper glacier surface and possibly greater ice velocities (Table 1). The total lidar–derived cumulative elevation change of the glacier surface in 2024 was –6.4, – 6.0, –5.3, and –5.5 m for Sites 1–4, respectively (Figure 5). Unfortunately, as previously discussed, the smart stakes did not provide reliable elevation data during early September—the warmest period of the 2024 season—since they were slowly

tipping over (Figure 5).

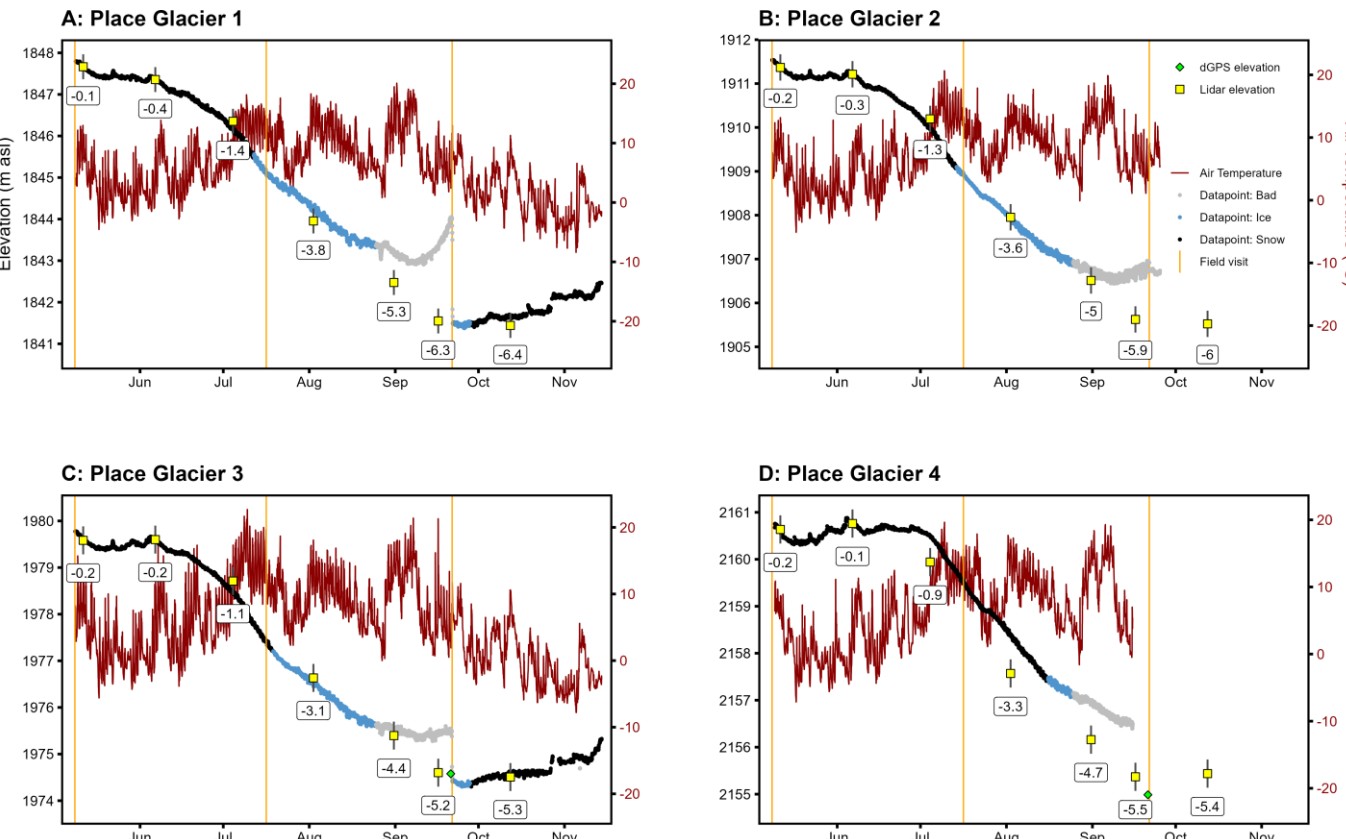

**Figure 5: Time series of hourly glacier elevation and air temperature from the four smart stakes (A–D). Glacier elevation points are colored as snow (black), ice (blue), and bad data points (grey). Independent elevation datasets are shown: lidar (yellow square with 30 cm error bars and labels of the cumulative elevation change in meters) and differential GPS (green diamond).**



## 4.4    Lapse rates

We combine five air temperature datasets to calculate monthly lapse rates for 2024 (Figure 6). They are –6.48, –6.48, –5.12, and –4.53°C km⁻¹ for the months of June through September, respectively. There is a non–linear lapse rate near the toe of the glacier in a cold micro–climate observed at Sites 1 and 2 that warms gradually to the plateau at Site 3 and then cools upwards to Site 4 following a near–normal lapse rate.

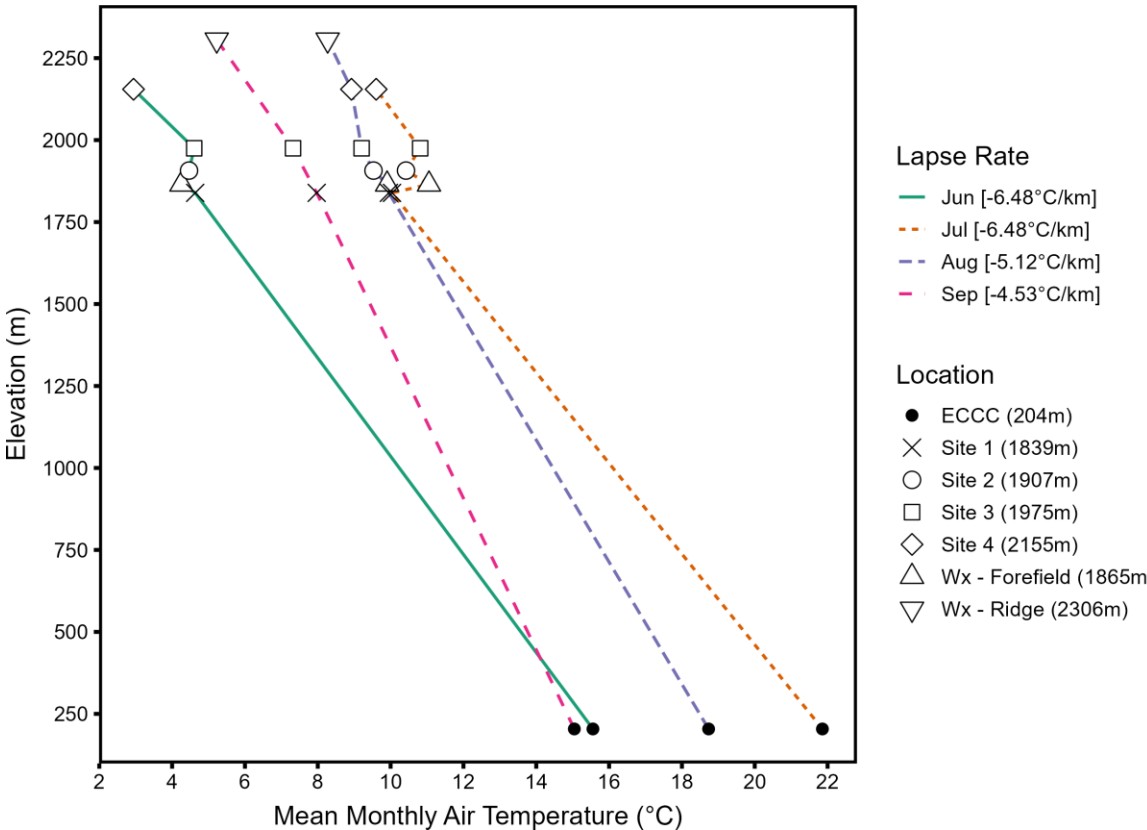

**Figure 6: Mean monthly air temperature lapse rates for June through September 2024. Linear monthly lapse rate coefficients are provided in the legend. Months with less than 95% complete data are omitted.**

## 4.5    Melt model

The ice melt factors ($k_i$) for Sites 1–4 are –5.02, –5.03, –4.26, and –5.63 mm w.e. °C⁻¹ d⁻¹, respectively (Figure 7), and the snow melt factors ($k_s$) are –4.38, –4.42, –3.78, and –3.74 mm w.e. °C⁻¹ d⁻¹. The average $k_i$ and $k_s$ are –4.99 and –4.08 mm w.e. °C⁻¹ d⁻¹, respectively. The R² values are greater than 0.98 for both ice and snow at all sites. Only the data from prior to the sensors tipping over are used in these calculations.





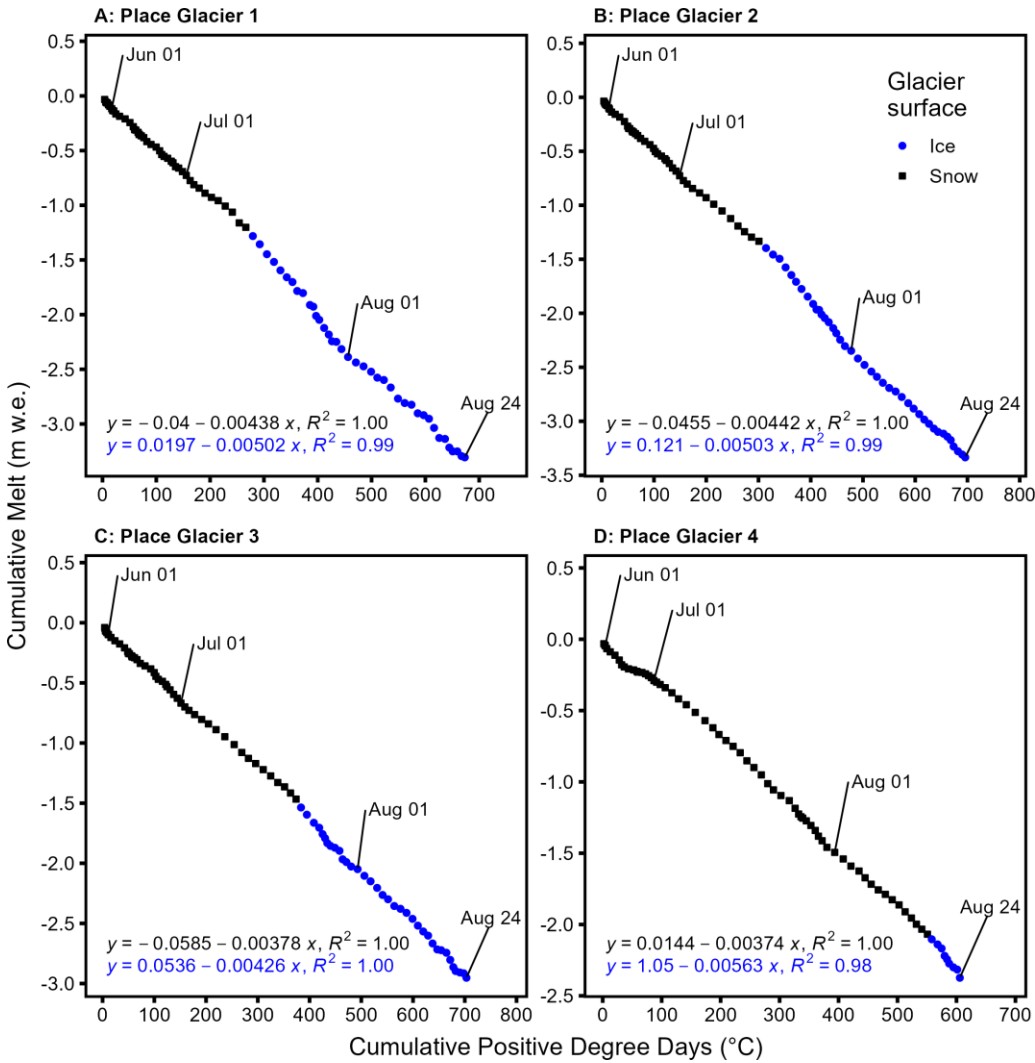

**Figure 7: Scatterplot of the cumulative positive degree days and the cumulative melt for each site (A–D). Points are colored by the**
**satellite derived glacier surface cover (snow or ice).**

The melt factors are then applied to the PDD from each site to derive surface melt (Figure 7). The coefficients are only applied
when air temperature is positive, and since the melt model does not account for accumulation, any accumulation in the
observational time series is added to the melt model result. The model RMSE is 0.09, 0.04, 0.09 and 0.11 m for Sites 1–4,
respectively (Figure 8). Overall, the melt model performs well at estimating the cumulative elevation change at our sites. Minor
variations between the model and the observations still exist, which may be caused by ice dynamics (e.g. velocity, emergence,
submergence), over–simplification of the melt–model (e.g. wind, albedo, rain, may also play a role), or due to field related
challenges (e.g. compaction of the snow at the station, tilting of the instruments that was not detected, or potentially the
crossarm turning on itself in the wind).





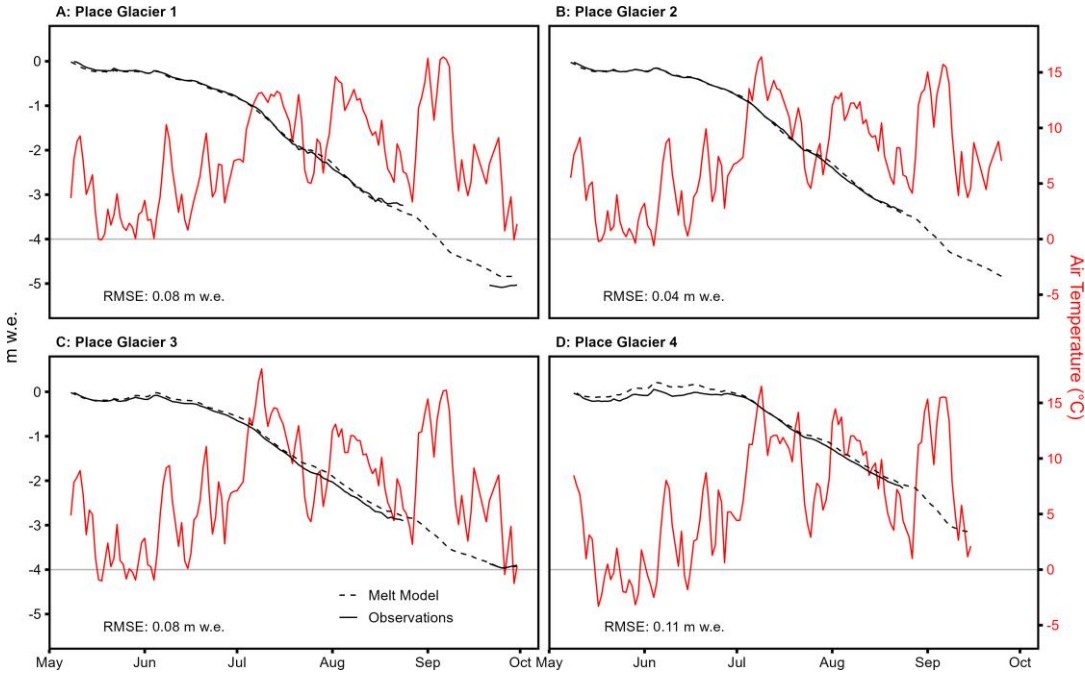

**Figure 8: Comparison of the observed melt from the smart stakes and the melt model. The air temperature is also shown for each station. The RMSE (in m w.e.) between the model and the observations is shown for each station.**

## 4.6    Spatial melt model

The total estimated seasonal melt volume from Place Glacier between May 14 and September 21, 2024, was 11.61 x 10$^6$ m$^3$ of water, or a summer balance of –4.14 m w.e. This date range covers the majority of the melt season, and has complete data from the lidar, air temperature and satellite snow cover maps. This estimate is derived from interpolating the daily PDD surface for the glacier and estimating the snow and ice extents and applying the average observed melt factors for ice and snow of –4.99 and –4.08 m w.e. °C$^{-1}$ d$^{-1}$, respectively (Figure 7). From this stack, we calculate the total PDD, total snow cover days, and the total melt (Figure 9).

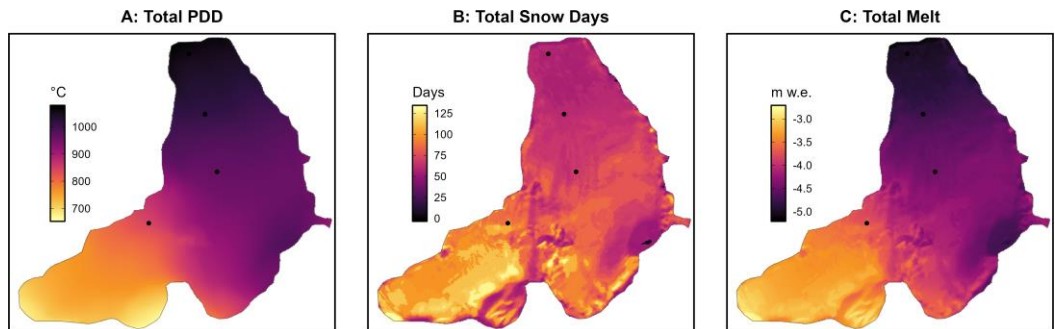

**Figure 9: Summary spatial melt–model from May 14 to September 21, 2024. A) Total PDD (°C), B) total snow days, and C) total melt (m w.e.).**





## 4.7    Melt model evaluation

We evaluated the spatial melt model against the manual ablation stakes from the long–term monitoring program at Place Glacier (Figure 1, Table 1, Table 2). The stakes are not used in the model. The stakes were drilled into the glacier ice at the end of the melt season on October 7, 2023. A handheld GPS was used to navigate within ~5 meters of the buried stakes on April 19–20, 2024 and the accumulated snow depth and density was recorded. On September 21, 2024, the stakes were surveyed, allowing for the calculation of the ice melt. The summer mass balance at the stakes—from April 19–20, 2024 to September 21, 2024—represents the total water equivalent of the accumulated snow and the ice melt (Table 2). The average summer mass balance from the manual ablation stakes is –3.42 m w.e.

**Table 2: Table of manually measured accumulated snow from October 7, 2023 to April 19, 2024, ice melt from October 7, 2023 to September 21, 2024. The net summer balance (total snow and ice melt).**

| Stake | Accumulated Snow (2023–2024) | | Ice Melt (2024) | | Summer Balance (2024) |
|---|---|---|---|---|---|
| | m | m w.e. | m | m w.e. | m w.e. |
| *30 | 2.26 | 1.06 | –4.49 | –4.04 | –5.10 |
| *35 | 1.79 | 0.84 | –3.21 | –2.89 | –3.73 |
| 40 | 2.50 | 1.18 | –3.36 | –3.02 | –4.20 |
| *44 | 3.31 | 1.40 | –2.87 | –2.58 | –3.98 |
| 45 | 2.34 | 0.98 | –2.81 | –2.53 | –3.51 |
| 50 | 2.50 | 1.06 | –2.14 | –1.93 | –2.99 |
| 52 | 3.36 | 1.42 | –1.92 | –1.73 | –3.15 |
| *75 | 3.69 | 1.56 | –1.73 | –1.56 | –3.12 |
| 80 | 4.25 | 1.80 | –2.61 | –2.35 | –4.15 |
| 90 | 3.61 | 1.53 | –1.34 | –1.21 | –2.74 |
| 95 | 3.67 | 1.55 | –1.02 | –0.92 | –2.4 7 |
| 100 | 3.12 | 1.32 | –1.49 | –1.34 | –2.66 |
| 120 | 3.27 | 1.38 | –1.46 | –1.31 | –2.69 |
| **Average** | **3.05** | **1.31** | **–2.34** | **–2.10** | **–3.42** |
| * Indicates the Spring 2024 field visit was on April 20, 2024. | | | | | |

These points are used as an independent validation dataset of the spatial melt model presented earlier. The main challenge in using the manual ablation stakes for the validation of the model is that the timing of the season does not coincide perfectly. The model is only run for the duration of the smart stake program because it depends on the smart stake derived air temperatures (May 14, 2024 to September 21, 2024), whereas the manual ablation stakes provide a summer mass balance from April 20, 2024 to September 21, 2024. Nevertheless, the melt model has an $R^2$ of 0.58 and an RMSE of 0.45 (Figure 10).




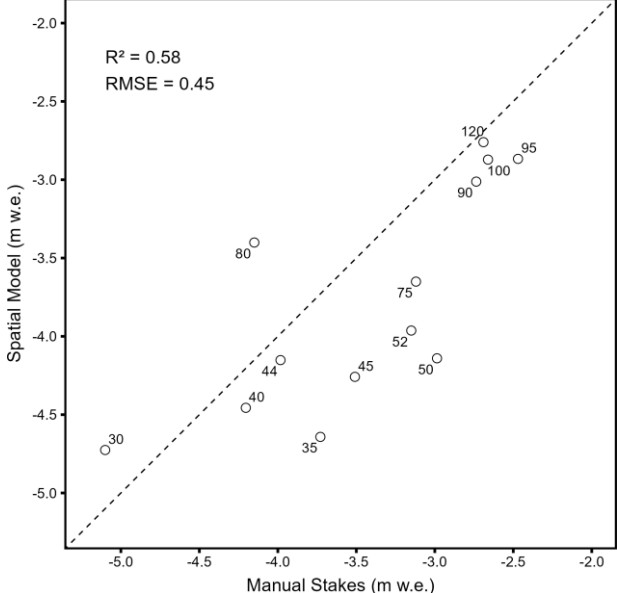

**Figure 10: Scatterplot of the summer mass balance at the manual stake locations from the manual measurements and from the spatial model, in meters water equivalent (m w.e.). The point labels are the stake names from Table 2, and the $R^2$ and RMSE (in m w.e.) are shown in the plot. The dashed line is a 1:1 line.**

### 4.8 Event monitoring

The melt model daily change rates are primarily negative during the ablation season with an overall average daily melt of –34, –38, –28, and –33 mm w.e. d$^{-1}$ (Figure 11). The majority of the melting occurred in the months of July, August, and September, with an average [maximum] daily melt rate of –47 [–82], –47 [–79], –38 [–68], and –41[–87] mm w.e. d$^{-1}$ at Sites 1–4, respectively.

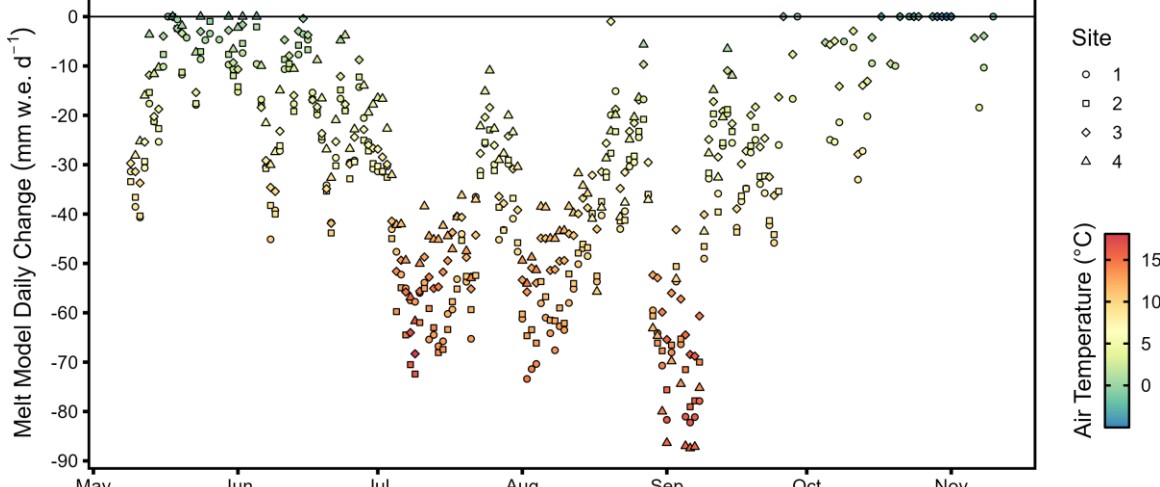

**Figure 11: Melt–model daily changes (mm w.e. d$^{-1}$) colored by average daily air temperature (℃).**



We observe three multi–day events where average air temperatures exceeded 10°C at all sites (Figure 12). The first event (Event A) occurred July 5 – 22, 2024, Event B took place August 1 – 12, 2024, and Event C happened August 29 – September 9, 2024. We compare the total melt from these three heat events to the total melt of the season. These three events account for on average 21.8, 14.7, and 18.3% of total summer melt, respectively. These events are responsible for over half of the total melt at Sites 1–3 (51.6, 52.2, 52.8%, respectively) and 62.5% of the total melt at Site 4 (Figure 12). Event A impacted all sites with between –0.82 and –1.08 m w.e. of melt in 17 days (average rate of –5.9 cm d⁻¹); Event B impacted all sites with between –0.54 and –0.77 m w.e. of melt in 11 days (–7.0 cm d⁻¹); and Event C impacted all sites with between –0.69 and –0.87 m w.e. of melt in 11 days (–7.6 cm d⁻¹), with an especially high melt rate at Site 4.

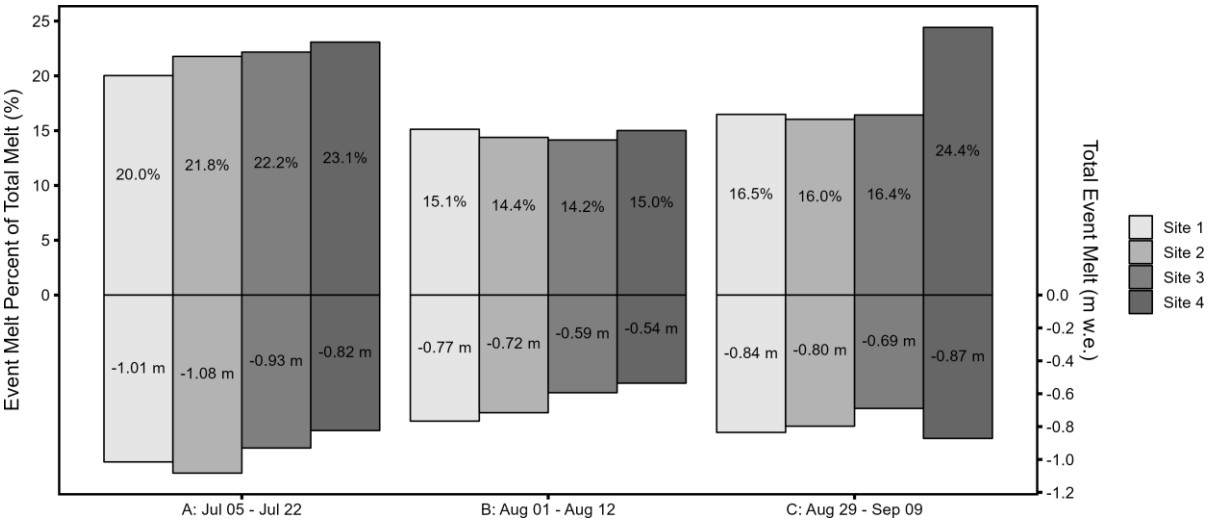

**Figure 12: Percent of seasonal melt for each station that occurred during melt events A) July 5–22, 2024; B) August 1–12, 2024; and C) August 29–September 9, 2024.**

## 5    Discussion

### 5.1    Practical considerations

The integration of low–cost sensors, Arduino microcontrollers, and satellite telemetry enabled automated collection of melt data throughout the ablation season at Place Glacier, British Columbia, Canada. Several practical advantages emerge from near real–time melt data (Cremona et al., 2023; Wickert et al., 2023). For example, automatic data backups to a remote server, network status dashboards to inform field logistics, and near real–time data for decision making and modelling (Northern BC Hydrology Research, 2025). In addition, their inexpensive nature is ideal for deployments on dynamic glacier surfaces where equipment loss is a risk (e.g. equipment falling into a crevasse; Table S1).

Smart stakes are not a substitute for fieldwork. In fact, given the substantial glacier melt observed globally in recent years (Hugonnet et al., 2021), more frequent field visits may be necessary to reset stakes before they melt out—especially near the glacier terminus (e.g., Figure 3). The smart stakes can operate anywhere with Iridium satellite coverage, though further testing



is needed for extreme environments. For example, the LiPo battery power management in high latitude or altitude environments with extended period of darkness or extreme cold has not been tested. In addition, the long–term viability of these units is not currently known.

## 5.2    Event–scale observations

There is a growing need to better understand high altitude extreme melt events and subsequent flooding, such as the 2021 heat
dome event in western North America (Reyes and Kramer, 2023). Traditional mass balance data is generally insufficient for melt attribution studies that require a dense time series of melt data (Kaspari et al., 2015). Our smart stake data quantified the significant impact of short–duration heat events on surface melt. In our study, more than half of the total seasonal melt occurred in only about a third of the ablation season due to three discrete warm events (Figure 11), and the warmest melt events had a greater impact at higher elevations (Figure 12). These findings align with Reyes and Kramer (2023), who documented
accelerated snowmelt during successive heat wave events in western North America. Event–scale observations of melt from smart stakes will allow future investigations of melt attribution from wildfires or snow–algae, for example (Bertoncini et al., 2022; Williamson and Menounos, 2021) and could one day inform improved operational hydrological forecasting (Kinnard et al., 2022; Lang, 1986).

Multiple sites across an elevation gradient helped us better understand glacier response during melt events. Site 4, for example,
experienced the greatest melt rate of all four sites, despite being the highest elevation (–87 mm w.e. d$^{-1}$). This event occurred during a less pronounced air temperature lapse rate (Figure S3). This highlights the importance of accurate air temperature models in glaciated environments (Ayala et al., 2015).

## 5.3    Temperature–index modeling

The melt factors we derived for ice (–4.26 to –5.63 mm w.e. °C$^{-1}$ d$^{-1}$) and snow (–3.74 to –4.42 mm w.e. °C$^{-1}$ d$^{-1}$) are similar
to values reported in previous studies (Braithwaite and Zhang, 2000; Carenzo et al., 2009; Shea et al., 2009; Wickert et al., 2023). For example, Shea et al. (2009) reported $k_i$ of 4.69 and $k_s$ of 2.71 mm w.e. °C$^{-1}$ d$^{-1}$ for Place Glacier, whereas Wickert et al. (2023) found a range of melt factors from 3.9 to 10.3 mm w.e. °C$^{-1}$ d$^{-1}$ across multiple sites from Antarctica to Alaska. The variation across study sites highlights the importance of local factors. We observed non–linear lapse rates across the glacier, with Sites 1 and 2 near the toe experiencing a cooler microclimate than would be expected from a linear lapse rate
(Figure 6). The off glacier weather station in the glacier forefield is generally warmer than the on–glacier sites, particularly in summer, even though it is 23 m higher in elevation than Site 1. These differences may be explained by the topography of the basin and katabatic wind flows (Ayala et al., 2015; Munro and Marosz-Wantuch, 2009). This spatial variability in temperature regimes underscores the importance of distributed temperature measurements across glaciers for accurate melt modeling, or perhaps the incorporation of satellite derived surface temperature datasets (Gök et al., 2024). A critical limitation of
temperature–index models is that they lump sensible heat flux and net radiation together, which can obscure important physical



processes driving melt (Landmann et al., 2021). The observed variability in the cumulative melt plots likely indicate non–stable melt factors throughout the season and highlight the need for energy balance approaches in future work (Figure 7).

The integration of in–situ data with airborne lidar and satellite observations demonstrates the power of multi–scale monitoring approaches (Cremona et al., 2023; Pelto et al., 2019). This hybrid approach creates a more comprehensive picture of glacier melt at different spatial and temporal scales (Figure 5), for example: 1) Smart stakes provide continuous, high–temporal resolution point measurements; 2) Repeat airborne lidar surveys provide high–precision spatial coverage of elevation change; and 3) Satellite data informs on the snow cover extent. The validation of smart stake measurements against independent lidar observations showed good agreement (RMSEs of $0.18 - 0.12$ m for Sites 1–3), though Site 4's higher RMSE (0.55 m) highlights the importance of considering local topography and ice dynamics when interpreting point measurements (Beedle et al., 2014).

## 5.4 Limitations of the study

**Smart stakes:** Several sources of uncertainty affect the accuracy of our measurements and melt estimates. For example, we could not correct the diurnal fluctuation in the ultrasonic data caused by the solar heating of the sensor (Figure S2). Correcting this bias requires the raw time–of–flight data, which the sensor does not record. The sensor, however, does have the ability to use an external temperature sensor to correct the speed of sound, which we will implement in the future. The fieldwork protocol of continuing the timeseries after the stakes tipped (e.g. Figure 5) over is a potential source of error. We were able to correct the data using secondary elevation change datasets (e.g. lidar and dGPS), which may not be the case in future deployments. Even though this study contributes notable advancements for glacier monitoring, snow and ice density data is still lacking, particularly over time. We used previously published snow densities from Pelto et al. (2019), however future work could have more frequent density measurements, a snow mass sensor (e.g. snow pillow), or combine remotely sensed snow density characterization (Darychuk et al., 2023).

**Glacier dynamics:** The emergence and submergence velocities of the glacier can cause vertical displacement of the glacier surface independent of ablation. Without direct measurements of ice velocity and flow patterns, these effects are difficult to quantify but likely contribute to the observed differences between modeled and lidar surface elevation changes. One solution in future work would be to combine this analysis with a better understanding of these fluxes. For example, Site 4 has anomalies that require further explanation regarding its low RMSE against the lidar elevations. With a slope of 16° this site is likely experiencing higher ice velocities which could be accounted for with better ice velocity data.

**Temperature–index model:** While temperature–index models offer a practical and computationally efficient approach to estimating snow and ice melt based on air temperature, they substantially oversimplify the underlying physical processes of energy exchange. These models neglect critical components of the surface energy balance—such as solar radiation variability, longwave radiation, wind-driven turbulent fluxes, and precipitation phase transitions—which can significantly influence melt dynamics, especially in heterogeneous mountain environments (Hock, 2003). Moreover, they assume a fixed relationship between temperature and melt (via the degree-day factor), yet this relationship varies with factors like albedo, cloud cover,



and elevation, leading to inaccurate melt estimates over space and time (Landmann et al., 2021; Walter et al., 2005; Wickert et al., 2023).

**Spatial melt model:** The spatial melt model does not account for non–linear changes in air temperature lapse rates and does not account for changing snow and ice densities over time. The snow and ice classification interpolates the snow cover during data gaps. In fact, a 90,000 $m^2$ supraglacial lake formed in 2024 on the surface of Place Glacier and drained in a glacier lake outburst flood on July 22, 2024 (Menounos et al., 2024). The lake location is evident in the south–southeast corner of the spatial melt model presented in this study (Figure 9). The lake location is erroneously classified as ice in the satellite image analysis (Figure S6). Furthermore, the digital elevation model is not representing the ice surface, but rather the lake surface, which has consequences for the interpolated daily elevation models and air temperature estimates (Figure S4, Figure S5). Consequently, the model is likely to overestimate the amount of melt in the location of the supraglacial lake as it assumes more ice cover. Since ice has a higher melt factor than snow, this error impacts the total melt volume reported in this analysis.

**Manual mass balance data:** The net mass balance reported by the World Glacier Monitoring Service (WGMS) has several limitations. Typically, the glacier is visited twice per year—once in spring and once in autumn. During the spring visit, stake readings are not possible, which means that any melt occurring after the late September visit is not captured within the proper hydrological year. Instead, this late–season melt is attributed to the following year. This discrepancy is primarily due to a combination of high ice melt rates, thick coastal snowpacks, and financial constraints that limit the frequency of site visits. As a result, the comparison of our melt estimates to the manual ablation stakes is not expected to be a 1-1 relationship (Figure 10).

## 5.5 Future smart stake development

Future development of smart stakes can benefit from several technological and methodological enhancements. In terms of logger construction and user interface, a key improvement would be the fabrication of custom Printed Circuit Boards (PCBs), which would facilitate faster assembly and standardized construction. The current reliance on the TPL 5110 for power management has proven problematic due to inconsistent wake-up times; alternatives should be explored. Additionally, building an Arduino library could streamline code management, while the creation of a graphical user interface (GUI) would improve device configuration and monitoring in the field.

Sensor integration is another area of potential advancement. Incorporating external temperature compensation could improve the accuracy of ultrasonic sensors. Secondary time-of-flight sensors, such as laser rangefinders, would add redundancy and enhance reliability. Integrating a compass, inclinometer, and/or accelerometer would allow for automated detection of stake tilt and orientation. Position tracking could be enabled through low-cost GPS or RTK GPS modules. Furthermore, adding sensors for wind speed, precipitation, and solar radiation would expand the climatic variables captured by the stake, enabling more comprehensive environmental monitoring.





For communication and power, implementing low-frequency radio communication between stakes and a central hub could significantly reduce telemetry costs. Exploring alternative satellite telemetry options, such as the RockBLOCK 9704 modem or other satellite constellations, would further enhance connectivity. Adaptive sampling strategies based on observed melt rates and battery voltage would optimize power usage. Designing compatibility with a 12 V battery bank could reduce reliance on solar power, increasing the system's robustness in variable weather conditions.

Physical design and field deployment protocols also warrant refinement. Developing alternative stake structures or anchoring methods could improve long-term stability under challenging glacial conditions. For example, a collar that stabilizes the pole and lowers with the snow surface could prevent tilting. Enhanced shielding for sensors would reduce bias from solar radiation. Field procedures should include regular differential GPS (dGPS) surveys to accurately track glacier surface elevation changes. Additionally, frequent snow density sampling would provide valuable calibration data for melt models.

Finally, improvements in data processing and analysis are essential to maximizing the utility of smart stake systems. Automated quality assurance and control routines would help maintain data integrity. The development of machine learning algorithms could uncover patterns in high-resolution melt data. Real-time alert systems for extreme melt events would offer critical insights for early-warning applications. Accounting for stake movement in data correction routines and processing spatial melt patterns in near real-time with each new data point would enhance both the accuracy and responsiveness of the system.

## 6  Conclusion

This research successfully developed and tested smart stakes on a relatively simple and slow–moving glacier, demonstrating their potential for advancing glacier monitoring with rich time series data. Smart stakes represent an important advancement that enables near real–time, high–temporal resolution ablation measurements critical for the current and future needs of glacier monitoring, flood forecasting, and hydrological modelling (Landmann et al., 2021).

The smart stake network at Place Glacier successfully captured high–resolution melt and meteorological data throughout the 2024 ablation season. While Sites 1 and 3 operated without issues, intermittent communication and sensor failures at Sites 2 and 4 highlighted challenges associated with remote monitoring in harsh alpine environments with low–cost equipment. Despite these setbacks, the data recorded on local storage allowed for a nearly complete seasonal analysis. The MB7374 ultrasonic sensors performed well overall, with a small RMSE when compared to *in–situ* measurements, though they were susceptible to solar heating effects. The satellite–derived snow cover observations effectively complemented *in–situ* measurements, helping define the onset of the snow–free season and improve the accuracy of the temperature–index melt model.

The spatial and time–series melt model results indicate that a few short periods of extreme heat contributed disproportionately to the total glacier mass loss. The three major warm events accounted for over half of the total seasonal melt, highlighting the sensitivity of glaciers to high–temperature anomalies, particularly at high elevations. While the melt–model performed well



overall, discrepancies between modeled and observed melt suggest that additional factors—such as ice dynamics, albedo variations, and wind effects—may influence surface melt rates beyond simple temperature–index relationships. Moving forward, refinements to the instrumentation, such as improved temperature compensation for ultrasonic sensors and additional redundancy in satellite communication, could enhance data reliability. These findings emphasize the importance of continuous

monitoring and improved modeling approaches to better understand the impact of climate variability on glacier melt processes.



**Code availability**

https://github.com/bevingtona/RemoteLogger/tree/main/v0.4_2024/general_purpose_ultrasonic_trh_TPL

**Author contribution:**

AB and BM developed the study and executed the field deployments. AB designed and built the smart stakes and developed
the code. ME collected the ablation stakes and retrieved the smart staked in September. AB prepared the manuscript with major
contributions from BM and minor contributions from ME.

**Competing interests:**

The authors declare that they have no conflict of interest.

**Acknowledgements**

We thank Hunter Gleason for his assistance with early prototyping of the Arduino data loggers. We acknowledge PlanetScope
for providing access to satellite imagery through an educational license granted to B. Menounos. We are also grateful to the
Arduino community for their contributions to open–source technology. Field support from Ben Pelto, Jeff Crompton and Bill
Floyd is gratefully acknowledged.

**Financial support**

Financial support for this research was provided by the British Columbia Ministry of Forests (Reference No. R&S-28), an
NSERC Canada Graduate Scholarship awarded to Bevington (Reference No. 518294), NSERC Discovery Grants awarded to
Menounos, and funding from the Tula Foundation.

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
