# Peer review of "Satellite telemetry of surface ablation observations to inform spatial melt modelling, Place Glacier, British Columbia, Canada"

_EGUsphere, 2025_

## Referee Comment (RC1)

**Review of Bevington et al. "Satellite telemetry of surface ablation observations to inform spatial melt modelling, Place Glacier, British Columbia, Canada", submitted to the Cryosphere.**

The submitted manuscript presents a novel setup of automated 'smart stakes' that transmit temperature, humidity and ultrasonic surface height data via Iridium satellite telemetry from the Place Glacier in Canada. The manuscript presents results from a summer campaign during 2024 finding a range of melt factors for snow and ice, which is variable based upon the unique temperature records stemming from each of the four smart stakes. The study finds that such networks can be highly valuable to improve attribution of melt to specific warm events with a near-realtime transmission of data. The authors provide a very detailed and thorough description of the logistical and technical setup of the smart stakes which is well written and beneficial to the community and to other scientists looking for ideas/inspiration for similar monitoring on glaciers. The study is generally interesting and the structure, writing and figures are very well put together. The authors should also be commended for the hard work that has gone into establishing and maintaining such a nice network.  While the quality of the work is mostly good, I am left questioning what is learned from some of the scientific findings in several places and I think that some of the data could and should be much better leveraged to inform a more detailed model. I believe that my comments compose a major revision to the manuscript before it should be published in the journal.

**Major Comments**

- The authors describe their framework as "comprehensive for understanding glacier melt dynamics", but their modelling approach is incredibly simplified, to the author's own admission. What is unclear is why the authors have not attempted to include a more detailed modelling approach based upon energy balance (if such data are available from the Ridge or Forefield AWS is not made clear), or even an enhanced temperature index model (e.g. Pellicciotti et al., 2005) that includes solar radiation (again data dependent from the off-glacier data). At the very least, the exclusion of accumulation (L323) when some level of height change is interpretable from the sonic rangers (with uncertainties) is perplexing. Even for a summer balance, such storm events, which can be critical for the surface albedo and energy budget, are important to include. Without that, I feel that the overall findings from the model are not overly informative or valuable, which does a disservice to the incredible data collection made by these 'smart stakes'. This is especially the case as the authors use high spatio-temporal data about snow cover from PlanetScope satellites. I believe the authors should test some form of model that accounts for summer accumulation and ideally at least shortwave radiation. Even if the modelling framework cannot be completely revised due to data constraints, some testing of this with nearby station or reanalysis data (for example) should be used to identify shortcomings of the model in the context of this new sensor network and build a more robust discussion surrounding its value to this "understanding of melt dynamics", when such dynamics are informed by a single DDF per site.

- The authors combine a lot of information from both their smart stake network and satellite (PlanetScope) and airborne (LiDAR) information. Some of this information is

used to support or understand the absolute elevation changes seen from the sonic rangers over the summer. However, the authors do not provide much perspective on such a setup where no detailed and contemporaneous airborne LiDAR surveys are available (which is almost always the case in glaciological research). For example, can the authors simply report height change relative to the start of the stake setup, without absolute numbers constrained by the LiDAR? The use of dGPS is also introduced here, but not discussed or presented any further. Is the data from those dGPS surveys sufficient without the LiDAR information to gain the same understanding from the telemetered information? This needs some more discussion and mention when presenting this as a more generalisable approach (which one feels is trying to be generalised when reading the very nice set up details for the Arduino etc). Also related to this discussion is the comparison to the 'smart stake' set up of Cremona/Landmann et al, which are cited. How and why is this approach superior or worth the extra effort/challenges/cost?

- I am left a little confused about the implementation of the lapse rate to distribute the air temperatures across the glacier, as the details are a little vague and the stations utilised include an airport station some distance away and a mix of on- and off-glacier weather stations/smart stakes. The authors state (L418) ". This spatial variability in temperature regimes underscores the importance of distributed temperature measurements across glaciers for accurate melt modeling" referring to the differences of temperature off- and on-glacier, but provides an unclear reasoning for this with respect to modelling, given the fact that they do not explore the relevance of using point-measured temperatures (smart stakes) vs. interpolation / lapsed temperatures and its impact on model accuracy.
Therein lies another main issue with the manuscript for me: Do the observed air temperature records at 4 stakes significantly benefit our overall modelling ability (given the issues with model, as above) compared to, say, 3 stakes, 2 or even 1, or only the off-glacier AWS? Would other studies likely benefit from just one station in the ablation zone and one in the accumulation zone, halving their costs? The authors need to i) establish a clear method and rationale for their lapse rate choice (just an hourly variable, linear lapse rate with all off- and on-glacier stations - Fig. S3?) - but why also the airport station included?, and ii) leverage this interesting and novel information to discuss more the value it brings to understand physical processes (e.g. boundary layer temperatures - Ayala et al., 2015 which includes Place Glacier) and melt dynamics and improve modelling.

- The value of the independent validation stakes (the manual stake measurements - Fig. 10) is not made overly clear and raises questions about the model. My understanding from the text and figure is that the period of manual measurements is longer (by a month) than the model period (limited due to the smart stake temperature operation). However, the information from Fig 10 shows that the model is still largely over-estimating the summer mass balances (is more negative). It is not so clear why that is (I suspect it is an albedo issue on the lower glacier which is not considered by the degree day modelling), but it does not build confidence in the value provided by the smart stakes if one wants to understand the mass balance of the whole glacier. For example, is it *just* the model, or also the distribution of temperatures? Given the reasonable performance of the model at the stakes

themselves (Fig.8), it is possibly the latter. But how sensitive is the model to that choice of data usage and does it undermine the value of the stakes if we still cannot confidently interpolate/distribute the temperature data between them, even on a small glacier? The authors need to build a clearer argument for this and provide more discussion to support how much is model uncertainty and how much is forcing uncertainty. I appreciate that this is not necessarily a study about modelling, so an exhaustive investigation of this is not necessary, but reasonable discussion and reflection on this for the reader is certainly needed.

**Specific Comments**

L69: Please revise the second objective of the manuscript here to be more precise. How exactly are they being compared to the PlanetScope imagery for snow cover and how does that help the overall message of the work?

L94/95: Can the authors provide the final date ranges here that they use in the model study?

L98-101: The authors introduce these weather stations, but it's not made clear if the forefield and ridge stations are permanent, what they measure and with what sensors/intervals/uncertainties. The authors should briefly report this information and particularly in light of the information that they could provide to a more detailed model (major comments above).

L127: The reported wake up times are naturally very short, due to battery life considerations, as is well reported by the authors. The equilibrium response times of the sensor according to the manufacturer are up to 30 seconds for the 63% range, but did the authors make any tests of these values compared to a continuously logging temperature sensor at the other weather stations, for example? Are the sensors also comparing well when placed together? Some short report of this would be beneficial.

L109-175: This section is very detailed and informative. The authors have done a great job to provide all of the necessary logistical and technical considerations. The authors should, however, provide a short table to summarize some of the information about chips/boards and instruments used for their final setup.

L153: Can the authors show some comparison of these temperature/RH observations compared to the AWS sensors or another reference? Are the accuracy records only taken from the manufacturer? What about a comparison between smart stake sensors before installation?

L196-198: As mentioned in my major comment above, it is unclear why exactly it is necessary to provide the absolute height changes via comparison with the airborne LiDAR, especially if the sites were dGPS located at the time of installation and stake re-setting (L215). Perhaps I have missed something clear here, but the authors should state their reasoning for this, and, as I mention above, provide discussion for the cases where this data is not available.

L218-226: Did the authors test other available datasets that did not require a user licence (even if researcher access is available for Planet when applying)? What about using Landsat and deriving albedo (e.g. Naegeli et al., 2019) to aid the development of an enhanced temperature index model? The authors run a distributed model using a 5 m LiDAR and 3m snow cover map, but it is unclear how much additional benefit that very high resolution has.

L222: How many of the 73 scenes remain after removal of those with haze/partial coverage etc?

Section 2.4.1: As per the major comment, please consider revising the modelling strategy or adapting it to test against cases with a more detailed model.

L241: if spring densities are not considered, why mention them here?

L269: The authors claim that accumulations are false when the air temperature is above 0°C, but snowfalls are possible at temperatures a few degrees above zero (e.g. Jennings et al., 2018), particularly in humid environments, like that of the study site. The authors should elaborate on their confidence of these false values and at what temperatures they are typically occurring.

L306-309: This short section requires some additional information. As per my major comment above, it is not clear how the lapse rates are applied in the glacier-wide model (only monthly means as in Fig. 6 or hourly variable?) and why the authors also include the airport station in this calculation. What is the goal of using those stations and in such a way? Do the authors wish only to make use of off-glacier data (that is typical in mountain regions) to see the value of smart stakes equipped with T/RH sensors? Do they only want to use on-glacier data to say that air temperature relates to melting of the glacier? How often is the lapse rate non-linear? How well does 1-3 stations represent the air temperature variability of another station in a leave-one-out analysis? Does it matter to have 4 stations? What is meant by a "near-normal" lapse rate (L309)? Please expand this section to explain the approach better and justify it within the context of conducting glaciological research with these very nice smart stakes.

L306: In this section it should also be clarified if the authors make any adjustments given the variable height of the temperature sensors above the surface. As the stakes melt out, the sensors will increasingly become independent of the boundary layer temperature variability due to the density driven katabatic winds (under warm weather conditions - Ayala et al., 2015). They will also become less affected by sources of uncertainty due to high albedo and heating errors of the radiation shield. These factors should also be considered when evaluating sub-period variability in calculated snow and ice melt factors.

L314: Are these melt factors useful when comparing the smart stake stations? Station 4's ice melt factor is clearly much higher due to the exposure of ice during a warmer period of observation. How high are the melt factors for the other stations if considering the same period as the snow-free period at station 4? The authors mention the variable nature of these factors, which is well established, but by how much are the differences?

L316: The R2 of what? The authors refer to Fig. 7 again here? Please say it explicitly.

L322-323: It is unclear how the 'accumulation' is added into the model results and what this approach really tells us about the response of the glacier to warming (the melt dynamics that the authors mention). So the glacier is still melting, but the height of the surface is superimposed on that, and the melt factor (if seen by Planet images) is set to snow? Again, I find this a key limitation that does a disservice to the great work in creating and maintaining this network.

L337: Define 'stack' in this context.

Section 4.7: Please see my major comment regarding the mis-match of the validation stakes and the reasoning/discussion about this.

L371: Can the authors specify what % if the total observation period these melt % occur under? This might help to give a more clear context.

L387-389: Can the authors elaborate on what is learned from the use of these smart stakes compared to normal stake measurements, or smart stakes based upon cameras (i.e. Landmann et al. 2021)? Again, given the over-simplified model approach and my concerns about the mismatch of the model (Fig. 10), the true value of the author's setup is not so clear.

L406: What is a less pronounced lapse rate? Do the authors refer to a shallower lapse rate (where the rate of change in temperature with elevation is less?). If so, state this clearly and provide an average value for context. It is still not clear from my reading what causes the melt rates to be so high for S4. Is this because of an ice melt factor that is derived from a short period of ice exposure and warm temperatures?

L416-419: Given that Place Glacier was used in the creation/testing of two models of air temperature distribution (Shea and Moore, 2010; Ayala et al., 2015), it is surprising that there was no comparison of this in the manuscript or discussion of these approaches using the airport or ridge/forefield stations as forcing. I understand that it is not the main aim of the work to look at temperature distribution, but, as mentioned before, the main value of the observed temperatures, linked to these such statements, are not so clearly demonstrated.

L421: Why did the authors not test the variable melt factors? My above point again related to the very high melt factor for S4.

L423-424: As mentioned, the authors should also highlight the value of their smart stakes for cases where airborne LiDAR data are not available, and where visible imagery might be increasingly limited by cloud during the melt season (e.g. parts of the Himalaya / S-E Tibet).

L442: The authors made dGPS measurements when setting up and resetting the SmartStakes. Is there a reason that ice velocity measurements are still unavailable? Perhaps I have missed something here.

**Figures / Tables**

Table 1: The authors should add what data are measured at each station

Fig. 3: Please add the station numbers to these plots, just for clarity.

Fig. 4: Change the legend to read 'start of ice *exposure*'.

Fig. 6: I think that this figure would benefit more from excluding the ECCC (maybe the current version could stay in the SI?), and zooming into the glacier area. It would be more valuable to see what the lapse rates are just by fitting to the on-glacier smart stakes and also perhaps when using only the two off-glacier stations (assuming no on-glacier data).

Fig. 7: Specify in the caption whether these are derived from the model or from the observations.

Fig. 9: I would invert the colour scales in all subplots, as the PDD, snow and melt are more intuitive if the oranges and yellows represent larger melt values/temperatures.

Fig. 10: Please add a colour to the circle to show the mean snow duration or something that might help to interpret the reason for the mis-match and its cause (see major comment).

Fig. 11: I think it would be clearer to indicate each site with a different colour, rather than the temperatures, as this mostly replicates the y-axis melt ranges.

**Cited Works**

Ayala, A., Pellicciotti, F., & Shea, J. (2015). Modeling 2m air temperatures over mountain glaciers: Exploring the influence of katabatic cooling and external warming. *Journal of Geophysical Research: Atmospheres*, *120*, 1–19. https://doi.org/10.1002/2015JD023137.Received

Cremona, A., Huss, M., Landmann, J. M., Borner, J., & Farinotti, D. (2023). European heat waves 2022: contribution to extreme glacier melt in Switzerland inferred from automated ablation readings. *Cryosphere*, *17*(5), 1895–1912. https://doi.org/10.5194/tc-17-1895-2023

Jennings, K. S., Winchell, T. S., Livneh, B., & Molotch, N. P. (2018). Spatial variation of the rain-snow temperature threshold across the Northern Hemisphere. *Nature Communications*, *9*(1), 1–9. https://doi.org/10.1038/s41467-018-03629-7

Landmann, J. M., Künsch, H. R., Huss, M., Ogier, C., Kalisch, M., and Farinotti, D.: Assimilating near-real-time mass balance stake readings into a model ensemble using a particle filter, The Cryosphere, 15, 5017–5040, https://doi.org/10.5194/tc-15-5017-2021, 2021.

Naegeli, K., Huss, M., & Hoelzle, M. (2019). Change detection of bare-ice albedo in the Swiss Alps. *The Cryosphere*, *13*, 397–412. https://doi.org/https://doi.org/10.5194/tc-13-397-2019

Pellicciotti, Francesca., Brock, Ben. W., Strasser, Ulrich., Burlando, Paolo., Funk, Martin., & Corripio, Javier. G. (2005). An enhanced temperature-index glacier melt model including the shortwave radiation balance : development and testing for Haut Glacier d ' Arolla , Switzerland. *Journal of Glaciology*, *51*(175), 573–587.

Shea, J. M., & Moore, R. D. (2010). Prediction of spatially distributed regional-scale fields of air temperature and vapor pressure over mountain glaciers. *Journal of Geophysical Research*, *115*(D23), D23107. https://doi.org/10.1029/2010JD014351

---

## Author Comment (AC1)

*Dear RC1,*

*We thank you for providing thorough and constructive comments on our manuscript. They highlight important gaps in our workflow, analysis, and discussion. In response to these comments, we significantly revised the manuscript. Our response to each comment can be found below in italicized, blue-faced font.*

*The main changes to the manuscript are:*

1. ***Refined Objectives:*** *We improved and clarified the objectives and made sure to clearly link them into discussion and conclusion of the paper. The objectives of the paper are to: 1) Describe the design and performance of smart stakes in a data rich environment; 2) Combine smart stake and remotely sensed data to inform a simple distributed mass balance model; and 3) Demonstrate how real–time ablation data can be used to examine the role of individual events on ablation.*

2. ***Enhanced Temperature-Index Model:*** *We integrated the Enhanced Temperature-Index model (ETI), while retaining the original Temperature-Index model (TI) from our original submission, and ran each model with multiple air temperature models. The ETI allows us to add incoming shortwave radiation, albedo, and snowfall to the analysis.*

3. ***Remote Sensing Analysis:*** *We added Sentinel-2 and Landsat 8/9 satellite imagery from the Harmonized Landsat Sentinel (HLS) dataset for snow cover mapping on the glacier. This HLS data is free and publicly available and complements the PlanetScope data that we used previously. We also use the HLS data to calculate broadband albedo over the glacier in the revised modelling framework.*

4. ***Lapse Rate Analysis:*** *We expanded our analysis of air temperature lapse rates and options for estimating air temperature across the glacier. We now test different combinations of stations (e.g. on glacier, off glacier) and different regression formulas (e.g. linear, polynomial). The highest performing model is a linear daily lapse rate of air temperature using only the on-glacier weather stations.*

5. ***Addressing the Katabatic Boundary Layer:*** *We discuss the importance of the katabatic boundary layer, and how future work could implement a study design that accounts for katabatic effects. We stress, however, that our experimental design is not ideal to evaluate the impact of katabatic flows on temperature downscaling to the glacier survey (investigating this phenomenon was never our stated goal in the paper).*

6. ***Geodetic Mass Balance:*** *We added lidar derived geodetic mass balance as a second independent validation dataset for our model. In our initial submission, the model did not perform very well against the independent mass balance data from manual ablation stakes. We suspect this poor performance was due to differences in the start/end dates of our logger experiment and those of the WGMS stake measurement program. Our hypothesis is partly supported by the good agreement between the geodetic and modeled mass change. We now discuss this in both the results and discussion section of the revised paper.*

7. ***Financial Cost:*** *We clarified the "low-cost" argument of the smart stakes. We moved the description of the overall cost of the smart stakes from the supplementary materials, where it was more cumbersome to find, to the main body of the paper. This strengthens the rational of the "low-cost" aspect of the project.*

8. ***Workflow Clarification:*** *We improved the description of the overall workflow in the methods and added a flowchart to simplify the explanation of the workflow. The flowchart helps guide the reader.*

9. ***Further Discussion of Site 4:*** *The model does not perform well at Site 4. We expanded our discussion on why this may be – the most likely contributor to the poor performance is that the stake was only drilled into snow (not into ice, as the snow was too deep) in the spring of 2024 and the stake likely shifted, tipped, or settled over time.*

*This significant re-working of the paper strengthened the quality and defensibility of the science and the overall relevance of the work.*

*We would like to emphasize that our smart stakes remain, to our knowledge, the first low-cost and open-source solution to real-time ablation data using satellite telemetry.*

*To reduce the length of the manuscript resulting in the new additions, we moved the "Future Smart Stake Development" section to the supplement.*

*We thank you for your time and efforts reviewing our revised work.*

*Kind regards,*

*Alexandre Bevington*
*on behalf of the authors*

**Reviewer Comment 1 (RC1)**

The submitted manuscript presents a novel setup of automated 'smart stakes' that transmit temperature, humidity and ultrasonic surface height data via Iridium satellite telemetry from the Place Glacier in Canada. The manuscript presents results from a summer campaign during 2024 finding a range of melt factors for snow and ice, which is variable based upon the unique temperature records stemming from each of the four smart stakes. The study finds that such networks can be highly valuable to improve attribution of melt to specific warm events with a near-realtime transmission of data. The authors provide a very detailed and thorough description of the logistical and technical setup of the smart stakes which is well written and beneficial to the community and to other scientists looking for ideas/inspiration for similar monitoring on glaciers. The study is generally interesting and the structure, writing and figures are very well put together. The authors should also be commended for the hard work that has gone into establishing and maintaining such a nice network. While the quality of the work is mostly good, I am left questioning what is learned from some of the scientific findings in several places and I think that some of the data could and should be much better leveraged to inform a more detailed model. I believe that my comments compose a major revision to the manuscript before it should be published in the journal.

**Major Comments**

1. The authors describe their framework as "**comprehensive for understanding glacier melt dynamics**", but their modelling approach is incredibly simplified, to the author's own admission. What is unclear is why the authors have not attempted to include a more detailed modelling approach based upon energy balance (if such data are available from the Ridge or Forefield AWS is not made clear), or even an **enhanced temperature index model** (e.g. Pellicciotti et al., 2005) that includes solar radiation (again data dependent from the off-glacier data). At the very least, the exclusion of **accumulation** (L323) when some level of height change is interpretable from the sonic rangers (with uncertainties) is perplexing. Even for a summer balance, such storm events, which can be critical for the surface albedo and energy budget, are important to include. Without that, I feel that the overall findings from the model are not overly informative or valuable, which does a disservice to the incredible data collection made by these 'smart stakes'. This is especially the case as the authors use high spatio-temporal data about snow cover from PlanetScope satellites. I believe the authors should test some form of model that accounts for summer accumulation and ideally at least shortwave radiation. Even if the modelling framework cannot be completely revised due to data constraints, some testing of this with nearby station or reanalysis data (for example) should be used to identify shortcomings of the model in the context of this new sensor network and build a more robust discussion surrounding its value to this "understanding of melt dynamics", when such dynamics are informed by a single DDF per site.

*We thank Referee #1 for their valuable input into ways in which we could improve our paper and ensure that our research is most informative to the scientific community. We have removed statements such as "comprehensive for understanding glacier melt dynamics" and have focussed more closely on the real–time data, the modelling, and the heat event analysis.*

*The components of the stakes, their ability to generate observations in real-time and our intention to publicly release their design to the glaciological community was our paper's overall intention. We also thought it would be important to demonstrate how data generated from the stakes could be used to tune a simple glacier melt model.*

*We added the ETI model to the study. The implementation of the ETI model (Pellicciotti et al., 2005) includes shortwave radiation from re-analysis data, albedo from HLS, and includes accumulation. The changes are most notable in **4.3 Temperature-index models** (methods); **5.6 Model evaluation and selection** (results), and **6.3 Contributions to melt** (discussion).*

*We certainly can appreciate that our initial melt model (and even our revised model) is not cutting edge and as described below, we lack important physical variables which preclude our ability to accurately model daily melt. The primary advantage of our ablation stakes is their ability to generate estimates of surface melt for varied surfaces of a given glacier (e.g. snow vs. ice, clean vs. debris-covered ice) much more cheaply than a conventional AWS especially if equipped with an Iridium modem for real-time data transmission.*

*We now explicitly address those points in our revised paper. The major changes we made in our revised paper include:*

- *Addition of statements about our paper's primary motivation, the simple nature of the TI and ETI melt modeling used to evaluate the stake's performance, and additional details about how our sensors could be used by others to quantify melt over varied glacier surfaces in real-time. For example:*

  - ***L64-69:*** *"We report on the development and implementation of low–cost near real–time ablation stakes that utilize ultrasonic sensors to measure accumulation and ablation that communicate outside of cell service–herein referred to as 'smart stakes'. The objectives of the paper are to: 1) Describe the design and performance of smart stakes in a data rich environment; 2) Combine smart stake and remotely sensed data to inform a simple distributed mass balance model; and 3) Demonstrate how real–time ablation data can be used to examine the role of individual events on ablation."*

  - ***L600-606:*** *"The smart stakes also present a complimentary dataset to on–glacier AWS because of the low–cost and ease of installation. The gains over a single weather station from the increased spatial sampling include: 1) quantifying the spatial distribution of melt and melt factors over diverse glacier facies (e.g. debris covered ice, dirty ice, steep slopes, or shaded regions), and 2) a quantification of the spatial distribution of air temperature beyond a single point. These allow for the calibration of the ETI model driven by smart stake data and spatial data. The smart stakes were tested in a data rich environment; however, they are suitable for any glacier and would provide important data at a low–cost for regions without repeat high resolution DEMs and in regions with poor optical satellite imagery."*

- *We have added additional descriptions of the available meteorological data available for our study and describe why such data is not sufficient to model surface energy using a full energy balance approach (See updated summary of the parameters measured at each weather station in **Table 2**).*

  - *"In order to distribute our smart stake glacier melt observations to the entire glacier, we use a simple temperature–index models since we lack important observational data for an energy balance model (Beedle et al., 2014; Braithwaite and Zhang, 2000; Carenzo et al., 2009; Pellicciotti et al., 2005; Shea et al., 2009; Wickert et al., 2023)."* ***L290-292***

  - *"These results, however, do not consider the full energy balance due to limited input data, and the covariance between air temperature and incoming shortwave radiation does not allow a proper melt partitioning (Kinnard et al., 2022)."* ***L531-533***

- *We add to our analysis the publicly available optical satellite imagery from the Harmonized Landsat Sentinel surface reflectance dataset (see **4.2.4 Optical Satellite Imagery**). The combination of HLS and Planet provide a more complete coverage of the melt season. Furthermore, HLS allows for broadband albedo to be calculated, an important variable for the Enhanced Temperature Index model suggested by the reviewer.*

2. The authors combine a lot of information from both their smart stake network and satellite (PlanetScope) and airborne (LiDAR) information. Some of this information is used to support or understand the absolute elevation changes seen from the sonic rangers over the summer. However, the authors do not provide much perspective on such a setup where no detailed and contemporaneous airborne LiDAR surveys are available (which is almost always the case in glaciological research). For example, can the authors simply report height change relative to the start of the stake setup, without absolute numbers constrained by the LiDAR? The use of dGPS is also introduced here, but not discussed or presented any further. Is the data from those dGPS surveys sufficient without the LiDAR information to gain the same understanding from the telemetered information? This needs some more discussion and mention when presenting this as a more generalizable approach (which one feels is trying to be generalised when reading the very nice set up details for the Arduino etc.). Also related to this discussion is the comparison to the 'smart stake' set up of Cremona/Landmann et al, which are cited. How and why is this approach superior or worth the extra effort/challenges/cost?

*These are all valid points and, as the referee rightly points out, the use of an absolute datum (e.g. elevation values above sea level) calculated from independent elevation data (e.g. dGPS and laser altimetry) is not required for general deployment of smart stakes.*

*In our case, since we only have dGPS measurements from the end of the season for two of the four sites, we used the lidar to correct the ultrasonic data after the stakes tipped over. There is no other way to know what the surface elevation change at the stake locations was between when they started to tip over to when they were re-installed. We refer the referee to the availability of lidar and dGPS data in **Figure 7**.*

*Building on the referee comments, we add in the paper that smart stakes would be especially valuable in areas with no repeat laser altimetry data and poor optical remote sensing products (e.g. steep glaciers, remote glaciers, and glaciers with persistent cloud cover). See **L502-505**: "In future work, multiple dGPS surveys and snow density measurements are recommended, but not essential. Repeat lidar and satellite imagery, however, are not required for smart stake deployment. The repeat lidar was helpful in this study to quantify the melt while the stakes were tipping over, and remote sensing identifies the snow and ice cover. The smart stakes are well suited for glaciers with no repeat lidar and challenging remote sensing conditions (e.g. persistent clouds)."*

*In the context of this paper, our primary aim is to test the smart stakes, and as such have included lidar and optical remote sensing. We highlight their use in more challenging environments in the concluding remarks: **L604-606**: "The smart stakes were tested in a data rich environment; however, they are suitable for any glacier and would provide important data at a low–cost for regions without repeat high resolution DEMs and in regions with poor optical satellite imagery."*

*Our smart stakes are different from the instrumentation used by Cremona/Landman. Those authors used cameras to take pictures of ablation stakes from which they could calculate melt. Our smart stakes use ultrasonic sensors to measure the distance from the sensor to the glacier. The smart stakes have the addition of real-time satellite telemetry. The smart stakes resemble more closely the instrumentation from Wickert et al., which we were unaware of during the early development of our study. More importantly, the instruments from Wickert et al. do not have satellite telemetry. We*

*discuss similarities of these in the introduction (**L46-63**) and in the description of the smart stakes (**3.1 Sensors**).*

3. I am left a little confused about the implementation of the lapse rate to distribute the air temperatures across the glacier, as the details are a little vague and the stations utilised include an airport station some distance away and a mix of on- and off-glacier weather stations/smart stakes. The authors state (L418) ". This spatial variability in temperature regimes underscores the importance of distributed temperature measurements across glaciers for accurate melt modeling" referring to the differences of temperature off- and on-glacier, but provides an unclear reasoning for this with respect to modelling, given the fact that they do not explore the relevance of using point-measured temperatures (smart stakes) vs. interpolation / lapsed temperatures and its impact on model accuracy. Therein lies another main issue with the manuscript for me: Do the observed air temperature records at 4 stakes significantly benefit our overall modelling ability (given the issues with model, as above) compared to, say, 3 stakes, 2 or even 1, or only the off-glacier AWS? Would other studies likely benefit from just one station in the ablation zone and one in the accumulation zone, halving their costs? The authors need to

   o i) establish a clear method and rationale for their lapse rate choice (just an hourly variable, linear lapse rate with all off- and on-glacier stations - Fig. S3?) - but why also the airport station included? and

   o ii) leverage this interesting and novel information to discuss more the value it brings to understand physical processes (e.g. boundary layer temperatures - Ayala et al., 2015 which includes Place Glacier) and melt dynamics and improve modelling.

*We can appreciate many of the points raised by the referee here in terms of how we apply our melt model and what is required to drive it. We do stress, however, that a proper treatment of the boundary layer physics at Place Glacier requires a different experimental design and, unfortunately, we lack many of the important datasets (e.g. wind speed, temperature at multiple levels at a given point) required to advance our understanding of the boundary layer. Nevertheless, the referee provides many useful questions which allow us to dig a bit deeper into our analysis. We now test six different lapse rates using the seven in-situ air temperature datasets (Described in **4.3.2 Spatial model implementation**). The lapse rates are shown in **Sections 5.4** and **5.6**, and in **Figure 8**, and in the supplement in **Figure S6, Figure S7, Figure S8**. We also discuss these findings in **6.3 Contributions to melt**. We have also added*

   o ***L324-327**: "Previous work demonstrated the importance of understanding on– and off–glacier air temperatures, and the important local influence of the ice temperature and air flow dynamics within the katabatic boundary layer (Ayala et al., 2015; Shea et al., 2009; Shea and Moore, 2010). Our air temperature models do not account for effects occurring inside or outside of the katabatic boundary layer, as we do not have sufficient data to address this."*

   o ***L539-543**: "These differences may be explained by the topography of the basin and katabatic wind flows (Ayala et al., 2015; Munro and Marosz-Wantuch, 2009). This spatial variability in air temperature regimes underscores the importance of distributed temperature observations across glaciers for accurate melt modelling. A possible improvement would be to include non–adiabatic influences including the*

*distance along the glacier and proximity to the ice margins into a distributed melt model (Ayala et al., 2015; Greuell and Böhm, 1998)."*

- o **L567-572**: *"Air temperature: We did not adjust air temperature values based on their height above the ice. The smart stakes were installed May 8/9, and they melted out to about ~3 m above the ice on July 16, and were redrilled to ~1 m. They tipped over in August and were re-drilled on again September 21. The air temperature sensors remained at the same position on the stakes (at the top), and as such changed their relative height above the glacier surface as the season progressed. As the sensors were between 1 and 3 m above the surface of the glacier for the duration of the season, they remain within the katabatic layer of ~5 m as found in Ayala et al. (2015)."*

4. The value of the independent validation stakes (the manual stake measurements - Fig. 10) is not made overly clear and raises questions about the model. My understanding from the text and figure is that the period of manual measurements is longer (by a month) than the model period (limited due to the smart stake temperature operation). However, the information from Fig 10 shows that the model is still largely over-estimating the summer mass balances (is more negative). It is not so clear why that is (I suspect it is an albedo issue on the lower glacier which is not considered by the degree day modelling), but it does not build confidence in the value provided by the smart stakes if one wants to understand the mass balance of the whole glacier. For example, is it just the model, or also the distribution of temperatures? Given the reasonable performance of the model at the stakes themselves (Fig.8), it is possibly the latter. But how sensitive is the model to that choice of data usage and does it undermine the value of the stakes if we still cannot confidently interpolate/distribute the temperature data between them, even on a small glacier? The authors need to build a clearer argument for this and provide more discussion to support how much is model uncertainty and how much is forcing uncertainty. I appreciate that this is not necessarily a study about modelling, so an exhaustive investigation of this is not necessary, but reasonable discussion and reflection on this for the reader is certainly needed.

*Thank you for being critical of this section. We summarize our response in the following four points:*

1. **Value of independent data**: *We chose to include the WGMS manual ablation stake data as a way to independently evaluate our model. Since the model aims to estimate the total melt over the ablation season, we opted to test it against a completely independent dataset that was not used in the training or calibration of the model.*

2. **Issues with manual stakes**: **L578-581** *"The mass balance reported by the WGMS from manual ablation stakes has several limitations. Typically, the glacier is visited twice per year—once in spring and once in autumn. During the spring, readings of the previous years' stake are not possible (buried), which means that any melt occurring after the late September visit is not captured within the proper hydrological year. Instead, this late–season melt is attributed to the following year."*

3. **Addition of a second validation dataset**: *Due to the poor performance of our model against the independent manual ablation stake data, we decided to evaluate our model*

*against a second independent dataset. We calculate the geodetic mass balance using the repeat lidar (see **5.6 Model evaluation and selection**, including **Figure 11** and **Figure 13**). This evaluation showed a notable improvement in model performance (from $R^2$=0.63 against the manual stakes to $R^2$=0.82). This suggests that the reviewer is correct in highlighting that the extra month of data from the manual stakes is likely causing the poor model performance. See the next point below.*

4. ***Timing discrepancy****: We clarified the discrepancy in measurement periods in their respective descriptions in the data section and discussed this in the Limitations (**Section 6.4**), see **L581-588**: "Furthermore, there is a discrepancy in the measurement periods for all of the input datasets. NRCan visited the manual stakes on April 19$^{th}$ and September 21$^{st}$, 2024. Whereas we installed the smart stakes on May 8$^{th}$, 2024, which had different end dates depending on their respective performances (**Figure 7**). The 2024 lidar flights were from May 11$^{th}$ to October 12$^{th}$. We ran the melt models from May 14$^{th}$ to September 21$^{st}$, 2024. Using May 14$^{th}$ as the beginning allowed for any initial settling of the smart stakes to occur. This discrepancy in measurement periods could explain a portion the higher performance of the model against the geodetic mass balance ($R^2$ = 0.82), which has similar dates to the smart stakes, compared to the performance against the manual stakes ($R^2$ = 0.63), which has an extra month of data in the spring."*

**Specific Comments**

L69: Please revise the second objective of the manuscript here to be more precise. How exactly are they being compared to the PlanetScope imagery for snow cover and how does that help the overall message of the work?

1) *Thanks for this comment. The PlanetScope data is simply used to distribute a melt model by knowing the snow vs. ice cover of the glacier over the season. We revised objective our objectives to clarify the focus of the paper: **L67-69** "1) Describe the design and performance of smart stakes in a data rich environment; 2) Combine smart stake and remotely sensed data to inform a simple distributed mass balance model; and 3) Demonstrate how real–time ablation data can be used to examine the role of individual events on ablation."*

L94/95: Can the authors provide the final date ranges here that they use in the model study?

*We added **L201**: "Smart stake data from May 8 to November 14, 2024, are used in this study."*

L98-101: The authors introduce these weather stations, but it's not made clear if the forefield and ridge stations are permanent, what they measure and with what sensors/intervals/uncertainties. The authors should briefly report this information and particularly in light of the information that they could provide to a more detailed model (major comments above).

*A fair point. We now mention in the text that these stations record observations longer than a season. We also added columns in **Table 2** for the measured parameters, sampling rate, date range, and network name and data access.*

L127: The reported wake up times are naturally very short, due to battery life considerations, as is well reported by the authors. The equilibrium response times of the sensor according to the

manufacturer are up to 30 seconds for the 63% range, but did the authors make any tests of these values compared to a continuously logging temperature sensor at the other weather stations, for example? Are the sensors also comparing well when placed together? Some short report of this would be beneficial.

*We apologize about this confusion. The battery consumption is a scenario with assumptions on timing. We reworded to add clarity. In reality, the startup time is about 10 seconds, and the Temp/RH measurement takes about 3 seconds, and the ultrasonic is sampled 10 times and the median value is reported. These points are now included in section **3.1 Sensors**.*

L109-175: This section is very detailed and informative. The authors have done a great job to provide all of the necessary logistical and technical considerations. The authors should, however, provide a short table to summarize some of the information about chips/boards and instruments used for their final setup.

*Thank you for this comment. We had supplied Table S1 in the supplement but now move this into the main as **Table 1**.*

L153: Can the authors show some comparison of these temperature/RH observations compared to the AWS sensors or another reference? Are the accuracy records only taken from the manufacturer? What about a comparison between smart stake sensors before installation?

*Added "from the manufacturer" (**L152**) and added **Figure S2**, a correlation plot between all of the air temperature sensors in the study. See **L151-154**: "Typical reported accuracies from the manufacturer are ±2%RH and ±0.3°C. The accuracy decreases to ±1.3°C at the limits of the temperature range. We found that the performance between the air temperature recorded at each smart stake deployed on the glacier had Pearson Correlation values greater than 0.96 between the observed daily air temperatures (Figure S2)."*

L196-198: As mentioned in my major comment above, it is unclear why exactly it is necessary to provide the absolute height changes via comparison with the airborne LiDAR, especially if the sites were dGPS located at the time of installation and stake re-setting (L215). Perhaps I have missed something clear here, but the authors should state their reasoning for this, and, as I mention above, provide discussion for the cases where this data is not available.

*Thanks for the comment. You are correct, the absolute height changes are not necessary. In our case, however, since we only have dGPS measurements from the end of the season for two of the four sites, we used the lidar to correct the ultrasonic data after the stakes tipped over. Otherwise, we are unable to know the surface elevation change at the stake locations between when they started to tip and when they were re-drilled. Note the availability of lidar and dGPS data in **Figure 7**. See a more detailed explanation in response to **RC1 Major Comment #2**.*

*We also emphasize in the conclusion: **L604-606**"The smart stakes were tested in a data rich environment; however, they are suitable for any glacier and would provide important data at a low–cost for regions without repeat high resolution DEMs and in regions with poor optical satellite imagery."*

L218-226: Did the authors test other available datasets that did not require a user licence (even if researcher access is available for Planet when applying)? What about using Landsat and deriving albedo (e.g. Naegeli et al., 2019) to aid the development of an enhanced temperature index model? The authors run a distributed model using a 5 m LiDAR and 3m snow cover map, but it is unclear how much additional benefit that very high resolution has.

*The reviewer raises a fair point and although this was a 'minor comment', we decided to test our workflow using publicly-available data. We used the Harmonized Landsat -Sentinel (HLS) surface reflectance data to map snow and ice and also to derive broadband albedo through time. Please refer to section **4.2.4 Optical Satellite Imagery**; and **5.2 Satellite derived snow cover**.*

L222: How many of the 73 scenes remain after removal of those with haze/partial coverage etc?

*A fair question. Since we now include the HLS data as well as PlanetScope, we clarified for all sensors: "We manually filtered out partial or cloudy images, leaving 15 Landsat, 25 Sentinel-2, and 31 PlanetScope Dove images, for a total of 71 images. These images represent 47 unique dates due to acquisitions on the same day by different sensors." **L264-266***

Section 2.4.1: As per the major comment, please consider revising the modelling strategy or adapting it to test against cases with a more detailed model.

*The modelling strategy is updated (along with the description in the methods, the results and the discussion). See response above to **RC1 Major Comment #1**.*

L241: if spring densities are not considered, why mention them here?

*Good point, removed.*

L269: The authors claim that accumulations are false when the air temperature is above 0°C, but snowfalls are possible at temperatures a few degrees above zero (e.g. Jennings et al., 2018), particularly in humid environments, like that of the study site. The authors should elaborate on their confidence of these false values and at what temperatures they are typically occurring.

*Thank you for this comment. We clarified that the perceived accumulation was during periods where air temperatures were >10°C: **L346-348** "As a precaution, we remove data from late–August to September 21 when the stakes recorded significant accumulation during a period of air temperatures greater than 10°C, when in reality the perceived accumulation was caused by the glacier surface becoming closer to the sensor while the stake was tipping over (Figure 5, Figure 7)."*

L306-309: This short section requires some additional information. As per my major comment above, it is not clear how the lapse rates are applied in the glacier-wide model (only monthly means as in Fig. 6 or hourly variable?) and why the authors also include the airport station in this calculation. What is the goal of using those stations and in such a way? Do the authors wish only to make use of off-glacier data (that is typical in mountain regions) to see the value of smart stakes equipped with T/RH sensors? Do they only want to use on-glacier data to say that air temperature relates to melting of the glacier? How often is the lapse rate non-linear? How well does 1-3 stations represent the air temperature variability of another station in a leave-one-out analysis? Does it matter to have 4 stations? What is meant by a "near-normal" lapse rate (L309)? Please expand this

section to explain the approach better and justify it within the context of conducting glaciological research with these very nice smart stakes.

*The lapse rate analysis is significantly restructured to test these effects. We now test six lapse rate models that include different combinations of on and off glacier weather stations with linear and polynomial models. The top performing model uses all of the on-glacier stations. A single on-glacier station would not capture the complex lapse rates that occur on Place Glacier. See response above to **RC1 Major Comment #3**.*

*In addition:*

- *Added **Figure 8** that shows daily lapse rates, the frequency of inversions and the performance of six lapse rate models against observational data. These daily inversions that occur on the glacier would not be captured with a single air temperature location.*

- ***L495-497:** "The number of stakes required per glacier in future applications will depend on the research question, for example: 1) sample many glaciers with a single smart stake near the equilibrium line altitude; 2) achieve complete coverage of the glacier's elevation range; or 3) capture niche glacier surfaces."*

- *In **Figure S2**, we show that Sites 2 and 3, and Site 4 and Wx-Ridge have strong correlations (0.99 and 0.98 respectively). We suspect that future placement on even more varied surfaces (e.g. feeling effects of a nearby nunatak, rock - turbulent transfer, or debris-covered ice) could resulted improved observations of processes involved in sensible heat transfer.*

L306: In this section it should also be clarified if the authors make any adjustments given the variable height of the temperature sensors above the surface. As the stakes melt out, the sensors will increasingly become independent of the boundary layer temperature variability due to the density driven katabatic winds (under warm weather conditions - Ayala et al., 2015). They will also become less affected by sources of uncertainty due to high albedo and heating errors of the radiation shield. These factors should also be considered when evaluating sub-period variability in calculated snow and ice melt factors.

*We added in our revised Limitations section:  **L567-572**: "Air temperature: We did not adjust air temperature values based on their height above the ice. The smart stakes were installed May 8/9, they melted out to about ~3 m above the ice on July 16, and were redrilled to ~1 m. They tipped over in August and were re-drilled on again September 21. The air temperature sensors remained at the same position on the stakes (at the top), and as such changed their relative height above the glacier surface as the season progressed. As the sensors were between 1 and 3 m above the surface of the glacier for the duration of the season, they remain are within the katabatic layer of ~5 m as found from model tuning in Ayala et al. (2015)."*

*Also, see the response to **RC1 Major Comment #3**.*

L314: Are these melt factors useful when comparing the smart stake stations? Station 4's ice melt factor is clearly much higher due to the exposure of ice during a warmer period of observation. How high are the melt factors for the other stations if considering the same period as the snow-free

period at station 4? The authors mention the variable nature of these factors, which is well established, but by how much are the differences?

*Thanks for the point. We now calculate the melt factors in three different ways. We also report the melt factors in Table 3.*

**L309-313:** *"We calculate TF and SRF in three different ways: 1) TF is calculated from daily PDD and daily melt values (TFDM); 2) TF is calculated from cumulative PDD and cumulative melt (TFCM); and 3) TF and SFR are calculated from solving a multiple linear regression from the daily values (TFMLR and SRFMLR). The two first methods are used in the TI model, Eq. (1), and the third method is used in the ETI model, Eq. (2). Melt factors are often similar among stakes, and are often assumed to be constant over time, however in reality they have been shown to change day–to–day, highlighting then need for energy balance models."*

L316: The R2 of what? The authors refer to Fig. 7 again here? Please say it explicitly.

*Thanks, good point. We corrected the caption: "**Figure 1**: Scatterplot of the observed cumulative positive degree days and the cumulative melt for each site (A–D). The coefficients for snow and ice are shown on the plot in m w.e. °C$^{-1}$ d$^{-1}$. Points are colored by the daily melt factors in mm w.e. °C$^{-1}$ d$^{-1}$. The regression formula and R$^2$ values are in grey for snow, and black text for ice."*

L322-323: It is unclear how the 'accumulation' is added into the model results and what this approach really tells us about the response of the glacier to warming (the melt dynamics that the authors mention). So the glacier is still melting, but the height of the surface is superimposed on that, and the melt factor (if seen by Planet images) is set to snow? Again, I find this a key limitation that does a disservice to the great work in creating and maintaining this network.

*Accumulation is added to the model from the ablation stake data. See **L330-331**: "Accumulation is accounted for in the model using the observed accumulation at each stake and interpolated across the glacier using elevation."*

L337: Define 'stack' in this context.

*Thanks for the comment. We changed the term to "Summary raster data from the daily ETI model using the on–glacier linear lapse rate reveal expected spatial variability in drivers of surface melt (Figure 12)" **L446-447**.*

Section 4.7: Please see my major comment regarding the mis-match of the validation stakes and the reasoning/discussion about this.

*Please see comments in regard to **RC1 Major Comment #4**.*

L371: Can the authors specify what % if the total observation period these melt % occur under? This might help to give a more clear context.

**L478-479:** *"We compare the total melt from these three events (39 days, 30% of total melt period) to the total melt of the season (130 days). These three events account for, on average, 23.5, 16.2, and 18.3% of total summer melt, respectively."*

L387-389: Can the authors elaborate on what is learned from the use of these smart stakes compared to normal stake measurements, or smart stakes based upon cameras (i.e. Landmann et al. 2021)? Again, given the over-simplified model approach and my concerns about the mismatch of the model (Fig. 10), the true value of the author's setup is not so clear.

*We highlight in multiple locations that the value of these stakes is clear, namely: 1) high temporal resolution; 2) Low cost; 3) Inclusion of multiple sensors (e.g. air temperature, RH, and suitable for expansion); 4) real-time data outside of cell service. Due to their low cost and relatively easy deployment, they are also suitable to sample multiple glacier facies*

***L590-592:*** *"We developed and tested inexpensive sensors (smart stakes) to monitor glacier melt using satellite communication. Smart stakes enable near real–time, high–temporal resolution ablation measurements critical for the current and future needs of glacier monitoring, flood forecasting, and hydrological modelling (Landmann et al., 2021)."*

***L600-603:*** *"The smart stakes also present a complimentary dataset to on–glacier AWS because of the low–cost and ease of installation. The gains over a single weather station from the increased spatial sampling include: 1) quantifying the spatial distribution of melt and melt factors over diverse glacier facies (e.g. debris covered ice, dirty ice, steep slopes, or shaded regions), and 2) a quantification of the spatial distribution of air temperature beyond a single point."*

L406: What is a less pronounced lapse rate? Do the authors refer to a shallower lapse rate (where the rate of change in temperature with elevation is less?). If so, state this clearly and provide an average value for context.

*The reference to a "pronounced lapse" rate now removed. We updated the lapse rate analysis and include six different lapse rate calculations in the analysis. A visualization of the daily air temperature lapse rates is now available in **Figure 8**. Also see response to **RC1 Major Comment #3**.*

It is still not clear from my reading what causes the melt rates to be so high for S4. Is this because of an ice melt factor that is derived from a short period of ice exposure and warm temperatures?

*We updated the modelling framework and results with the Enhanced temperature index model. The new model uses our updated lapse rate analysis (using the on-glacier stations with a daily linear lapse rate) improved the performance of the model. The total melt during the three events decreases with elevation, see updated **Figure 14**. The total amount of melt from the heat events is lowest at Site 4, but the proportion of the total melt of the season from the heat events is highest at site 4.*

L416-419: Given that Place Glacier was used in the creation/testing of two models of air temperature distribution (Shea and Moore, 2010; Ayala et al., 2015), it is surprising that there was no comparison of this in the manuscript or discussion of these approaches using the airport or ridge/forefield stations as forcing. I understand that it is not the main aim of the work to look at temperature distribution, but, as mentioned before, the main value of the observed temperatures, linked to these such statements, are not so clearly demonstrated.

*Thank you for this comment. We have added "Previous work demonstrated the importance of understanding on– and off–glacier air temperatures, and the important local influence of the ice temperature and air flow dynamics within the katabatic boundary layer (Ayala et al., 2015; Shea et al., 2009; Shea and Moore, 2010). Our air temperature models do not account for effects occurring inside or outside of the katabatic boundary layer, as we do not have sufficient data to address this."* **L324-32**

*We also now test six different air temperature lapse rate models, including on forced only with the airport station (see* **Figure 8,** *and* **Figures S7 and S8***).*

*We also added a section in the uncertainty of the model.* **L540-543** *"This spatial variability in air temperature regimes underscores the importance of distributed temperature observations across glaciers for accurate melt modelling. A possible improvement would be to include non–adiabatic influences including the distance along the glacier and proximity to the ice margins into a distributed melt model (Ayala et al., 2015; Greuell and Böhm, 1998)"*

L421: Why did the authors not test the variable melt factors? My above point again related to the very high melt factor for S4.

*We updated the modelling framework with the ETI model and tested three ways of deriving melt factors. See response to major comment above. We also show the variability of daily melt factors in* **Figure 9**.

L423-424: As mentioned, the authors should also highlight the value of their smart stakes for cases where airborne LiDAR data are not available, and where visible imagery might be increasingly limited by cloud during the melt season (e.g. parts of the Himalaya / S-E Tibet).

*As we described in previous comments and in the revised paper, the lidar data is not essential to the smart stakes. We simply used the lidar data as a way to test the smart stake deployment and use of a melt model. We believe that these smart stakes will find most utility in regions without additional geodetic data and where real time estimates of surface mass change are required. We added a concluding remark to this effect here* **L604-606** *("The smart stakes were tested in a data rich environment; however, they are suitable for any glacier and would provide important data at a low–cost for regions without repeat high resolution DEMs and in regions with poor optical satellite imagery."). We also emphasize that the smart stakes provide the ability to observe melt rates over varies glacier surfaces.*

L442: The authors made dGPS measurements when setting up and resetting the SmartStakes. Is there a reason that ice velocity measurements are still unavailable? Perhaps I have missed something here.

*Thanks for this comment. We apologise that it was not made clearer. We only had one date with successful dGPS measurements at two sites (see* **Section 4.2.3 dGPS***;* **Figure 7***; and multiple mentions in the* **Discussion***).*

Figures / Tables

Table 1: The authors should add what data are measured at each station

*Done. See revised **Table 2**.*

Fig. 3: Please add the station numbers to these plots, just for clarity.

*Done. See new **Figure 5**.*

Fig. 4: Change the legend to read 'start of ice exposure'.

*Done. See new **Figure 6**.*

Fig. 6: I think that this figure would benefit more from excluding the ECCC (maybe the current version could stay in the SI?), and zooming into the glacier area. It would be more valuable to see what the lapse rates are just by fitting to the on-glacier smart stakes and also perhaps when using only the two off-glacier stations (assuming no on-glacier data).

*We updated the figure to test four linear lapse rates and two polynomial lapse rates. See new **Figure 8**.*

Fig. 7: Specify in the caption whether these are derived from the model or from the observations.

*Good point. Added "observed" in the caption. Now **Figure 9**.*

Fig. 9: I would invert the colour scales in all subplots, as the PDD, snow and melt are more intuitive if the oranges and yellows represent larger melt values/temperatures.

*We updated the figure (**Figure 12**). It now shows the total snow days, PDD, SR, Albedo, TF, SRF and Melt. We have made the colors more intuitive where reds and yellows indicate more melt, whereas blues indicate cooler / more snow.*

Fig. 10: Please add a colour to the circle to show the mean snow duration or something that might help to interpret the reason for the mis-match and its cause (see major comment).

*We added the site elevation as the size of the dots. We also add a comparison of the sites against the geodetic mass balance. Now **Figure 11**.*

Fig. 11: I think it would be clearer to indicate each site with a different colour, rather than the temperatures, as this mostly replicates the y-axis melt ranges.

*The purpose of Figure 11 (now **Figure 14A**) is to convey visually the three melt events. Associating the shape with the site and the color with air temperature provides a simple visualization of the daily melt, and the association with pronounced warmer air temperatures. We would like to keep **Figure 14A** as it is.*

Cited Works

Ayala, A., Pellicciotti, F., & Shea, J. (2015). Modeling 2m air temperatures over mountain glaciers: Exploring the influence of katabatic cooling and external warming. Journal of Geophysical Research: Atmospheres, 120, 1–19. https://doi.org/10.1002/2015JD023137.

Cremona, A., Huss, M., Landmann, J. M., Borner, J., & Farinotti, D. (2023). European heat waves 2022: contribution to extreme glacier melt in Switzerland inferred from automated ablation readings. Cryosphere, 17(5), 1895–1912. https://doi.org/10.5194/tc-17-1895-2023

Jennings, K. S., Winchell, T. S., Livneh, B., & Molotch, N. P. (2018). Spatial variation of the rain-snow temperature threshold across the Northern Hemisphere. Nature Communications, 9(1), 1–9. https://doi.org/10.1038/s41467-018-03629-7

Landmann, J. M., Künsch, H. R., Huss, M., Ogier, C., Kalisch, M., and Farinotti, D.: Assimilating near-real-time mass balance stake readings into a model ensemble using a particle filter, The Cryosphere, 15, 5017–5040, https://doi.org/10.5194/tc-15-5017-2021, 2021.

Naegeli, K., Huss, M., & Hoelzle, M. (2019). Change detection of bare-ice albedo in the Swiss Alps. The Cryosphere, 13, 397–412. https://doi.org/https://doi.org/10.5194/tc-13-397-2019

Pellicciotti, Francesca., Brock, Ben. W., Strasser, Ulrich., Burlando, Paolo., Funk, Martin., & Corripio, Javier. G. (2005). An enhanced temperature-index glacier melt model including the shortwave radiation balance : development and testing for Haut Glacier d ' Arolla , Switzerland. Journal of Glaciology, 51(175), 573–587.

Shea, J. M., & Moore, R. D. (2010). Prediction of spatially distributed regional-scale fields of air temperature and vapor pressure over mountain glaciers. Journal of Geophysical Research, 115(D23), D23107. **https://doi.org/10.1029/2010JD014351**

---

## Author Comment (AC2)

*Dear RC2,*

*We thank you for providing thorough and constructive comments on our manuscript. They highlight important gaps in our workflow, analysis, and discussion. In response to these comments, we significantly revised the manuscript. Our response to each comment can be found below in italicized, blue-faced font.*

*The main changes to the manuscript are:*

1. ***Refined Objectives:*** *We improved and clarified the objectives and made sure to clearly link them into discussion and conclusion of the paper. The objectives of the paper are to: 1) Describe the design and performance of smart stakes in a data rich environment; 2) Combine smart stake and remotely sensed data to inform a simple distributed mass balance model; and 3) Demonstrate how real–time ablation data can be used to examine the role of individual events on ablation.*

2. ***Enhanced Temperature-Index Model:*** *We integrated the Enhanced Temperature-Index model (ETI), while retaining the original Temperature-Index model (TI) from our original submission, and ran each model with multiple air temperature models. The ETI allows us to add incoming shortwave radiation, albedo, and snowfall to the analysis.*

3. ***Remote Sensing Analysis:*** *We added Sentinel-2 and Landsat 8/9 satellite imagery from the Harmonized Landsat Sentinel (HLS) dataset for snow cover mapping on the glacier. This HLS data is free and publicly available and complements the PlanetScope data that we used previously. We also use the HLS data to calculate broadband albedo over the glacier in the revised modelling framework.*

4. ***Lapse Rate Analysis:*** *We expanded our analysis of air temperature lapse rates and options for estimating air temperature across the glacier. We now test different combinations of stations (e.g. on glacier, off glacier) and different regression formulas (e.g. linear, polynomial). The highest performing model is a linear daily lapse rate of air temperature using only the on-glacier weather stations.*

5. ***Addressing the Katabatic Boundary Layer:*** *We discuss the importance of the katabatic boundary layer, and how future work could implement a study design that accounts for katabatic effects. We stress, however, that our experimental design is not ideal to evaluate the impact of katabatic flows on temperature downscaling to the glacier survey (investigating this phenomenon was never our stated goal in the paper).*

6. ***Geodetic Mass Balance:*** *We added lidar derived geodetic mass balance as a second independent validation dataset for our model. In our initial submission, the model did not perform very well against the independent mass balance data from manual ablation stakes. We suspect this poor performance was due to differences in the start/end dates of our logger experiment and those of the WGMS stake measurement program. Our hypothesis is partly supported by the good agreement between the geodetic and modeled mass change. We now discuss this in both the results and discussion section of the revised paper.*

7. ***Financial Cost:*** *We clarified the "low-cost" argument of the smart stakes. We moved the description of the overall cost of the smart stakes from the supplementary materials, where it was more cumbersome to find, to the main body of the paper. This strengthens the rational of the "low-cost" aspect of the project.*

8. ***Workflow Clarification:*** *We improved the description of the overall workflow in the methods and added a flowchart to simplify the explanation of the workflow. The flowchart helps guide the reader.*

9. ***Further Discussion of Site 4:*** *The model does not perform well at Site 4. We expanded our discussion on why this may be – the most likely contributor to the poor performance is that the stake was only drilled into snow (not into ice, as the snow was too deep) in the spring of 2024 and the stake likely shifted, tipped, or settled over time.*

*This significant re-working of the paper strengthened the quality and defensibility of the science and the overall relevance of the work.*

*We would like to emphasize that our smart stakes remain, to our knowledge, the first low-cost and open-source solution to real-time ablation data using satellite telemetry.*

*To reduce the length of the manuscript resulting in the new additions, we moved the "Future Smart Stake Development" section to the supplement.*

*We thank you for your time and efforts reviewing our revised work.*

*Kind regards,*

*Alexandre Bevington*
*on behalf of the authors*

**Reviewer Comment 2 (RC2)**

This manuscript describes the design and deployment of a novel "smart stake" system that monitors surface melt and air temperature in near real time. The authors apply the data collected to a simple spatial melt modelling framework and use them to evaluate the influence of heat waves on glacier melt at Place Glacier, British Columbia.

The manuscript presents a clear and detailed description of the smart stake design and deployment. The use of low-cost sensors and satellite telemetry is interesting and has the potential to make glaciological monitoring more accessible. The real-time transmission capability is a valuable feature, and the overall approach contributes to ongoing conversations about alternatives to traditional on-ice AWS.

While the smart stake concept is promising, I was not fully convinced of its added value relative to a conventional on-ice AWS setup, and I found the melt modelling and event attribution to be fairly simplistic. The manuscript covers many topics but could benefit from more depth in each of them. Addressing some of these limitations would be important before publication.

In summary, I enjoyed reading about the smart stake setup and seeing its performance and data outputs. However, I found the presentation of the subsequent analyses too simplistic. I detail these comments below.

*We thank the reviewer for their detailed review of our paper and providing concrete ways in which we can strengthen our paper. In light of their comments and those of Referee #1, we completed additional analysis to address the major criticisms of our paper. As suggested, we now provide an updated ablation model that incorporates the influence of short wave radiation in our distributed model of surface mass balance. We stress, however, that the primary motivation of our paper was to describe the smart skate design and their general performance. Since we lack many of the key observations (e.g. wind speed over the glacier surface), we refrain from employing a full energy balance model to estimate distributed daily ablation for the glacier. We now alert the reader to the paper's primary objectives in the paper's introduction.*

**Major comments**

Cost argument: The main stated benefit of the smart stakes is their low cost, but no cost assessment is provided in the manuscript. Including such an assessment would be very useful for evaluating this setup. Without an explicit comparison, the argument for "low-cost" deployment remains difficult to evaluate. Furthermore, several suggested improvements (e.g., adding sensors to address tipping or solar heating) could make the smart stakes nearly as complex as an on-ice AWS, which would further weaken the cost advantage.

*Thank you for the comment. In order to streamline the initial draft of our manuscript, we decided to include the cost of the components in the paper's supplement (Table S1). To better support our claim for the stake's low cost, we moved this table into the main (now Table 1). The main cost advantage is the data logger and telemetry.*

*We added: "The total overall cost of the smart stake is approximately $1,100 USD. Most of this cost is made up of the Iridium modem and the ultrasonic sensor (Table 1). This cost is only a fraction of typical costs for a real–time AWS, which are typically in the range from $10,000 to $20,000 USD."* (**L119-121**).

*The added suggestion for addressing the tipping issues could likely be addressed with an accelerometer/gyroscope (e.g. ~$ 13 USD here: https://www.adafruit.com/product/3886). And: "Similarly, an inexpensive tiltmeter could be added to the stake identify tilting." (**L501**)*

Increased spatial resolution: While the smart stakes did improve temporal resolution compared to seasonal mass balance surveys, the gain over a single on-ice AWS is less clear, particularly when combined with mass balance stake measurements. The poor performance of the upper site further reduces the usefulness of deploying four sites. Would a single smart stake or on-ice AWS at mid-elevation, combined with the off-glacier stations for lapse rates, provide comparable results? A stronger case for the sensors' value could be made by explicitly testing the added benefit relative to existing approaches, especially given the availability of two off-ice AWS at this site.

*Thanks for this comment. We agree that the gain over traditional mass balance stakes is clear, and that the argument that smart stakes are better than an on-glacier AWS is less clear. We would like to clarify that the smart stakes do not attempt to replace on-glacier AWS. Rather, they aim to provide a*

*cost-effective solution to increasing the sampling of surface elevation change and air temperature across the glacier across multiple glacier surfaces and elevations.*

*We added concluding remarks to this effect in: "The smart stakes also present a complimentary dataset to on–glacier AWS because of the low–cost and ease of installation. The gains over a single weather station from the increased spatial sampling include: 1) quantifying the spatial distribution of melt and melt factors over diverse glacier facies (e.g. debris covered ice, dirty ice, steep slopes, or shaded regions), and 2) a quantification of the spatial distribution of air temperature beyond a single point." **L600-603**.*

*The gain from increased spatial sampling is especially important in areas where there is no repeat lidar, or other means of acquiring high-resolution elevation change, and important for understanding spatial distribution of air temperature. These are things that a single AWS is not able to do, and typically multiple full AWS are not deployed across elevations on a single glacier. There is no on-glacier AWS on Place Glacier, and the performance of the smart stakes against a single AWS was not tested. We add: "The smart stakes were tested in a data rich environment; however, they are suitable for any glacier and would provide important data at a low–cost for regions without repeat high resolution DEMs and in regions with poor optical satellite imagery." **L604-606***

*The poor performance of the upper site (Site 4) against the lidar data, and against the model results could be due to ice dynamics or potentially could be explained by snow settling beneath the stake, as it is the only stake which was not drilled into ice. We stress, however, that the confounding factor of ice dynamics is only important in our case where we are evaluating elevation change of the stakes against the geodetic data.*

*We do not believe that this poor performance is an argument against smart stakes but rather emphasises that quantifying ice dynamics and the full energy balance is likely required in complex topography. See **L176-178**: "For Sites 1–3, the snowpack was thin enough during the initial installation to drill the poles into the underlying ice, but thick snow at Site 4 prevented drilling the poles into the underlying ice." And **L551-552**: "A likely explanation of the poor model performance at Site 4 could be that it is the only stake that was not drilled into glacier ice, it was only drilled into the snow and may have settled over time."*

***Supplementary Figures S7 and S8** show the performance of the melt models using multiple air temperature models. **Figure S6** shows the overall model performance of each of those combinations. This demonstrates that although the air temperature model selection is important, the model selection is more important, with the ETI model outperforming the TI model using any of the considered air temperature models.*

Spatial melt modelling: The modelling approach is quite rudimentary, applying uniform melt factors across snow and ice despite calculating melt factors at four individual stakes. As a result, it is not clear how the smart stake data meaningfully enhance the analysis. The model performs reasonably, but not particularly well, and the value added by the smart stakes is not evident.

*This is a fair criticism. As now emphasized in the introduction of the paper, it was not our intention to introduce a state-of-the-art physical model in this paper but to use a rudimentary melt model and show how it might be calibrated to obtain distributed melt during the ablation season. However,*

*we did take this criticism seriously and decided to explore the importance of shortwave radiation in melt. We replaced the simple temperature index model with an Enhanced Temperature-Index Model. This model is now run using variable temperature records (**Section 4.3, Figure S7-S8**). In addition, we test melt factors from daily data, cumulative data, and from a multiple linear regression (**Table 3**). We also investigate the variability of melt factor values (**Figure 9**).*

*As stated above, we remind readers that we are not proposing a new or better model but rather informing a simple melt model using a combination of remote sensing data and real-time observations.*

Heat wave analysis: Much of the main ablation season is classified as "heat waves," making it unsurprising that a large fraction of melt occurred during those periods. More detail on how heat waves were defined, and how these events compare with other years, would strengthen this section. As currently framed, the analysis feels shallow, particularly since it relies heavily on site 4, which performed poorly compared to the lidar. The paper might benefit from focusing more deeply on either the melt modelling or the heat wave analysis, rather than presenting both at a fairly simplified level.

*Thank you for this comment. We agree that the paper could focus on either the melt modelling, or the event-monitoring. In this case, due to the incomplete observational record (stakes tipped over), we require a melt model to complete the time series. That time series is then used for a simple assessment of event-scale monitoring across 4 locations over the glacier. We believe this is fully in the scope of a single paper and increases the strength and utility of the research.*

*That said, we do agree that the term heat wave is not well defined. We opt for the term "heat events". **L391-392:** "We recorded three heat events, herein defined as times when the mean daily air temperature was above 10°C (Figure 8A)."*

*Indeed, Site 4 performed poorly. We believe this is due to either: 1) unquantified drifting in of the field instrument (e.g. the sensor turned/pivoted), 2) it started tipping much earlier than the other sites as it was only drilled into snow, or 3) ice dynamics/velocity which are unaccounted for in our model. We believe that despite these challenges, this data and modelling exercise showcase high resolution, real-time applications of low-cost smart ablation stakes, which could support real-time melt modelling and event scale analysis in the future. Both of which are important steps forward in glacier monitoring.*

*We also added the following sections to address the performance at Site 4: "The validation of smart stake measurements against independent lidar observations showed good agreement (RMSEs of 0.18–0.12 m for Sites 1–3), though Site 4's higher RMSE (0.55 m) highlights the importance of considering installation conditions, local topography and ice velocity when interpreting point measurements (Beedle et al., 2014). A likely explanation of the poor model performance at Site 4 could be that it is the only stake that was not drilled into glacier ice, it was only drilled into the snow and may have settled over time." (**L548-552**)*

**Minor comments**

It was not clear to me why the ECCC station is included in the lapse rate calculation. This choice made the glacier sites appear bundled together and harder to interpret. Clarification would be helpful.

*We can appreciate why the reviewer was confused here. Our rationale to include this station was that it provided a low elevation site to estimate a lapse rate above the ice surface. To make it clear, we revised the air temperature lapse rate component of this paper. Our revisions include: 1) better description of the weather stations (**Table 2**), multiple lapse rate models tested (**Section 4.3.2 Spatial model implementation**), and a more thorough explanation of the lapse rate evaluation we employed (**Section 5.4 Lapse rates**).*

In several places, the writing alternates between very detailed and overly casual phrasing, which occasionally disrupted the flow. For example, line 99: "some 400 m away" . Could this be made more precise?

*We updated to: "The first, "Wx–Forefield", is a weather station run by NRCan located 412 m down valley from the glacier terminus located on bedrock in the glacier forefield. The second, "Wx–Ridge", is a new weather station installed in early summer 2024 on an alpine ridge above the glacier." (**L214-216**)*

When justifying sensor or method choices, referencing prior use is not always sufficient. For example, line 140 states that a method was "used in other glaciological studies." It would be stronger to explain whether it worked well in those studies and what was gained by its use.

*Thanks for the comment. We added more detail to what was done and what is recommended by Wickert et al. 2023: "It is specifically optimized for measuring snow and similar sensors have recently been used successfully by Wickert et al. (2023). The authors tested the sensor on multiple glaciers around the world, although without satellite telemetry, and provided useful recommendations for future field deployments that we were not aware of at the time of our installation. Wickert et al. (2023) recommend the MaxBotix MB7388, and also tested the MB7060, MB7389, and MB7386. A comprehensive comparison of the available sensors from MaxBotix was not done in this study." (**L127-131**).*

If the issue with using GOES is that the station might move and lose connection, could an option such as transmitting data by radio signal to the main off-ice station, and then using GOES, be feasible?

*Yes, this is a great idea and one in which we are actively exploring for future deployment. We now discuss the aspect in the third paragraph of Supplementary Section **Future smart stake development:** "For communication and power, implementing low-frequency radio communication between stakes and a central hub could significantly reduce telemetry costs (Denissova et al., 2025). Exploring alternative satellite telemetry options, such as the RockBLOCK 9704 modem or other satellite constellations, could further enhance connectivity (e.g. GOES)."*

---

## Author Comment (AC3)

*Dear Dr. Pelto,*

*We thank you for providing thorough and constructive comments on our manuscript. They highlight important gaps in our workflow, analysis, and discussion. In response to these comments, we significantly revised the manuscript. Our response to each comment can be found below in italicized, blue-faced font.*

*The main changes to the manuscript are:*

1. ***Refined Objectives:*** *We improved and clarified the objectives and made sure to clearly link them into discussion and conclusion of the paper. The objectives of the paper are to: 1) Describe the design and performance of smart stakes in a data rich environment; 2) Combine smart stake and remotely sensed data to inform a simple distributed mass balance model; and 3) Demonstrate how real–time ablation data can be used to examine the role of individual events on ablation.*

2. ***Enhanced Temperature-Index Model:*** *We integrated the Enhanced Temperature-Index model (ETI), while retaining the original Temperature-Index model (TI) from our original submission, and ran each model with multiple air temperature models. The ETI allows us to add incoming shortwave radiation, albedo, and snowfall to the analysis.*

3. ***Remote Sensing Analysis:*** *We added Sentinel-2 and Landsat 8/9 satellite imagery from the Harmonized Landsat Sentinel (HLS) dataset for snow cover mapping on the glacier. This HLS data is free and publicly available and complements the PlanetScope data that we used previously. We also use the HLS data to calculate broadband albedo over the glacier in the revised modelling framework.*

4. ***Lapse Rate Analysis:*** *We expanded our analysis of air temperature lapse rates and options for estimating air temperature across the glacier. We now test different combinations of stations (e.g. on glacier, off glacier) and different regression formulas (e.g. linear, polynomial). The highest performing model is a linear daily lapse rate of air temperature using only the on-glacier weather stations.*

5. ***Addressing the Katabatic Boundary Layer:*** *We discuss the importance of the katabatic boundary layer, and how future work could implement a study design that accounts for katabatic effects. We stress, however, that our experimental design is not ideal to evaluate the impact of katabatic flows on temperature downscaling to the glacier survey (investigating this phenomenon was never our stated goal in the paper).*

6. ***Geodetic Mass Balance:*** *We added lidar derived geodetic mass balance as a second independent validation dataset for our model. In our initial submission, the model did not perform very well against the independent mass balance data from manual ablation stakes. We suspect this poor performance was due to differences in the start/end dates of our logger experiment and those of the WGMS stake measurement program. Our hypothesis is partly supported by the good agreement between the geodetic and modeled mass change. We now discuss this in both the results and discussion section of the revised paper.*

7.  ***Financial Cost:*** *We clarified the "low-cost" argument of the smart stakes. We moved the description of the overall cost of the smart stakes from the supplementary materials, where it was more cumbersome to find, to the main body of the paper. This strengthens the rational of the "low-cost" aspect of the project.*

8.  ***Workflow Clarification:*** *We improved the description of the overall workflow in the methods and added a flowchart to simplify the explanation of the workflow. The flowchart helps guide the reader.*

9.  ***Further Discussion of Site 4:*** *The model does not perform well at Site 4. We expanded our discussion on why this may be – the most likely contributor to the poor performance is that the stake was only drilled into snow (not into ice, as the snow was too deep) in the spring of 2024 and the stake likely shifted, tipped, or settled over time.*

*This significant re-working of the paper strengthened the quality and defensibility of the science and the overall relevance of the work.*

*We would like to emphasize that our smart stakes remain, to our knowledge, the first low-cost and open-source solution to real-time ablation data using satellite telemetry.*

*To reduce the length of the manuscript resulting in the new additions, we moved the "Future Smart Stake Development" section to the supplement.*

*We thank you for your time and efforts reviewing our revised work.*

*Kind regards,*

*Alexandre Bevington*
*on behalf of the authors*

**Community Comment 1 (CC1)**

The authors have provided a detailed examination of the efficacy and operational approach to using smart stakes. Place Glacier, British Columbia is the location for this detailed field test. This location has a combination of long-term mass balance records, ongoing mass balance observations, automatic weather station, PlanetScope and recent annual Lidar observations. This makes for a perfect test site location. The authors emphasize repeatedly that smart stakes are low cost as a key part of this study but have not provided a cost range for the product or its operation. The cost is as essential as identifying their accuracy, for determining if they are best suited as a supplement to an existing stake network, adding high temporal resolution data or can be deployed in place of that network. There have been other detailed ablation surveys conducted during specific heat events that should be noted as part of this important growing data set. Smart stakes can be a valuable tool for expanding this data set. Melt models based on data with limited temporal resolution can be impaired by temporal and spatial variations in ice dynamics, albedo variations, and wind effects. The smart stakes can be effective in identifying the temporal variations that impact all ablation stake studies.

I applaud the authors for a thorough test of a new system and providing a best-case approach to utilization of extensive complimentary data sets for both analysis and validation. In future it would be wonderful to see how much melt context smart stake data can provide from a glacier that otherwise has limited field monitoring but has ongoing LIDAR or other geodetic observation.

*We thank Dr. Pelto for his community comments on our paper. We appreciate the enthusiasm around our efforts to produce a rich dataset from the smart stakes. The detailed costs have been moved from the supplementary materials to the main body of the paper in* **Table 1**.

**Specific Comments**

93: It is noted the upper part of the accumulation area was not instrumented due to logistical and safety reasons. However, earlier it was noted that the glacier has minimal crevassing. Is it worth being more specific on this constraint.

*Dr. Pelto raises a fair point. It is true that minimal crevassing lowers the safety risk of glacier travel, however there are still crevasses. We decided to not travel to the top of the glacier on the original installation date due to low visibility. We added: "The uppermost region of the accumulation area was not instrumented due to field safety and logistical constraints during the May 2024 fieldwork, namely poor visibility at higher elevations caused by low cloud."* **L198-200**

220: Other studies have utilized snow line migration across areas of previously measured snow depth to identify ablation. Is this what is being done with PlanetScope?

*The Planet data is now combined with Landsat and Sentinel to determine the surface cover of the glacier at the ablation stakes (either snow or ice). This facilitates the determination of snow vs ice melt factors and is used in the distributed melt model. We do not explicitly calculate the snowline elevation in this paper. See updated* **Section 4.2.4**.

269: This is substantial tipping, did the manual stakes suffer this level of tilt due to near melt out? Are the smart stakes not emplaced as deeply or are they simply top heavy and prone to tilt earlier? The 4.88 m long stakes were drilled how deeply, it is noted that at least 0.8 m is exposed?

*There are two sets of manual stakes at every manual stake site: one set is drilled deep and dropped into the glacier, the other (at the same locations), is drilled in less deep and is exposed above the glacier. This allows for substantial melt to occur and there is always one of the sets of stakes that is exposed. The manual mass balance stakes were conducted independently from the smart stakes, and these programs did not coordinate drill depths. In addition, the manual stakes are not top heavy.*

*We added a description of the tipping mechanism and some ideas around solutions in future deployments: "Based on our observations, the tipping over of the smart stakes arises from a feedback between heating of the aluminum pole and heat transfer to one ice edge along the pole: as the pole melts into the edge more of the pole is in contract with the ice thereby accelerating leaning of the pole. This effect could potentially be mitigated in the future with a longer stake that is drilled deeper in the glacier, or by developing a self–adjusting tripod. Similarly, an inexpensive tiltmeter could be added to the stake to identify tilting."* **(L597-501)**

286: Figure 4A provides an excellent visual of snowpack variation. I recommend that the accumulation area ratio be reported for each. Given the difference in melt rates for snow surface vs ice surface this is important.

*Thanks for the comment. We separated the figure into two: Figure 4 and Figure 6). We also added the AAR over time in* **Figure 6B**.

316: This similarity in ablation from stake to stake has been noted for other regional glaciers, which maybe worth noting that this is not unusual.

*Good point. Added a note in the methods: "Melt factors are often similar among stakes, and are often assumed to be constant over time, however in reality they have been shown to change day–to–day, highlighting then need for energy balance models." (***L312-313***)*

339: The number of melt days is a crucial variable to identify for a melt model to work accurately, the mapping of this variable in Figure 9 is quite valuable as a an example of best practice

*Thank you for this comment. We updated the figure (now* **Figure 12***) so it now includes: The total snow days; the total positive degree days (PDD) using the on-glacier linear lapse rate; the mean incoming shortwave radiation (SR); the total melt from the ETI model with on-glacier linear lapse rate; the mean albedo (α); the mean temperature melt factor (TF$_{MLR}$); and the mean shortwave radiation melt factor (SRF$_{MLR}$).*

400: Important to note the increased specific melt rates observed during 24 recent heat wave event in the Nooksack Basin, North Cascades (Pelto et al. 2022) that supports observations provided here.

*Agreed, added reference to this observation: "These findings align with Reyes and Kramer (2023), who documented accelerated snowmelt during successive heat wave events in western North America, and with Pelto et al. (2022), who observed melt rates increase during heat waves in the Nooksack Basin, North Cascades." (***L511-513***).*

412: Providing a better reference to more specific regional studies where melt factors were derived demonstrates contrast and context. Bidlake et al (2010) noted melt factors for South Cascade Glacier of 0.0039+ 0.0006 for snow and 0.0056 + 0.0008 for ice. On Mount Baker in the North Cascades, Pelto et al (2022) reported under overall weather conditions DDFs for snow is 0.0035 m w.e. °C−1d−1. For ice, the DDFi is 0.0053 m w.e. °C−1d−1 During heat waves this rose to a DDFs snow of 0.0043 m w.e. °C−1d−1. For ice, the DDFi is 0.0067 m w.e. °C−1d−1.

*We added these values into the discussion about melt factors: "For example, Shea et al. (2009) reported TFi of -4.69 and TFs of -2.71 mm w.e. °C $^{-1}$ d $^{-1}$ for Place Glacier, whereas Wickert et al. (2023) found a range of melt factors from -3.9 to -10.3 mm w.e. °C $^{-1}$ d $^{-1}$ across multiple sites from Antarctica to Alaska. Bidlake et al. (2010) noted melt factors for South Cascade Glacier of -3.9 mm w.e. °C $^{-1}$ d $^{-1}$ for snow and -5.6 mm w.e. °C $^{-1}$ d $^{-1}$ for ice. On Mount Baker in the North Cascades, Pelto et al. (2022) reported -3.5 mm w.e. °C $^{-1}$ d $^{-1}$ for snow and -5.3 mm w.e. °C $^{-1}$ d $^{-1}$ for ice, and that these rose during heat waves to -4.3 and -6.7 mm w.e. °C $^{-1}$ d $^{-1}$, respectively." (***L524-528***)*

425: The four smart stakes did provide high resolution temporal data but at a low spatial resolution raising again the cost issue.

*Thanks for this comment. The costs were originally presented in the supplementary materials (Sup. Table 1), which has now been moved to the Main, in* **Table 1**. *The total cost is ~$1100 USD.*

442: Good description of the challenges posed by the vertical component of velocity. Make sure to note that this poses the same challenge for any ablation stake system . Smart stakes may in fact allow for better understanding of this.

*Excellent point. "Surface mass balance stakes, automatic or manual, do not account for the horizontal and vertical components of glacier dynamics (Beedle et al., 2014)."* **L565-566**

References

Bidlake, W.R., Josberger, E.G. and Savoca, M.E.: Modelled and Measured Glacier Change and Related Glacioloical, Hydrological and Meterorological conditions at South Cascade Glacier, WA, Balance and water years 2006-2007. USGS Science Investigation Report 2010-5143, US Geological Survey, Reston VA USA, 2010.

Pelto, M. S., Dryak, M., Pelto, J., Matthews, T., & Perry, L. B.: Contribution of Glacier Runoff during Heat Waves in the Nooksack River Basin USA. Water (7), 1145. https://doi.org/10.3390/w14071145, 2022.